

# Reliability-based design optimization of offshore wind turbine support structures using analytical sensitivities and factorized uncertainty modeling

Lars Einar S. Stieng[1] and Michael Muskulus[1]

[1]Department of Civil and Environmental Engineering, Norwegian University of Science and Technology NTNU, Trondheim, Norway

**Correspondence:** Lars Einar S. Stieng (lars.stieng@ntnu.no)

**Abstract.** The need for cost effective support structure designs for offshore wind turbines has led to continued interest in the development of design optimization methods. So far, almost no studies have considered the effect of uncertainty, and hence probabilistic constraints, on the support structure design optimization problem. In this work, we present a general methodology that implements recent developments in gradient-based design optimization, in particular the use of analytical gradients, within the context of reliability-based design optimization methods. By an assumed factorization of the uncertain response into a design-independent, probabilistic part and a design-dependent, but completely deterministic part, it is possible to computationally decouple the reliability analysis from the design optimization. Furthermore, this decoupling makes no further assumption about the functional nature of the stochastic response, meaning that high fidelity surrogate modeling through Gaussian process regression of the probabilistic part can be performed while using analytical gradient-based methods for the design optimization. We apply this methodology to several different cases based around a uniform cantilever beam and the OC3 Monopile and different loading and constraints scenarios. The results demonstrate the viability of the approach in terms of obtaining reliable, optimal support structure designs and furthermore show that in practice only a limited amount of additional computational effort is required compared to deterministic design optimization. While there are some limitations in the applied cases, and some further refinement might be necessary for applications to high fidelity design scenarios, the demonstrated capabilities of the proposed methodology show that efficient reliability-based optimization for offshore wind turbine support structures is feasible.

## 1 Introduction

Offshore wind energy is becoming an increasingly competitive alternative to the traditional land-based wind farms. However, there remains a level of additional cost which, together with some practical challenges, ensure that offshore wind is still a secondary consideration in many markets. Hence, the reduction of this cost is a primary objective in current research and development. Cost reduction is generally a multidisciplinary issue, including turbine components like rotor blades and the drivetrain, wind farm layout, the electrical grid and the design of the support structure (including the tower). Methods to derive cost effective, *optimal* support structure designs – balancing minimal use of materials (and potentially other cost driving design





aspects) with the ability to safely withstand the loads required by design standards – has been an active area of research for many years. However, very few studies have taken into account the probabilistic, fundamentally uncertain, aspects of the design process. This includes, for example, uncertainties in the environment and the modeling of the environment, affecting the loads experienced by the structure, as well as uncertainties about the details of the design itself, affecting the response to the applied

loads. Taking such uncertainties into account generally requires the use of probabilistic mathematical methods that severely complicate the design optimization problem that needs to be solved, both formally and numerically. Hence, deterministic safety factors tend to be used. This is also true even for single design assessments. The present study aims to address these issues by proposing a methodology that allows the use of both state-of-the-art optimization methods recently developed for support structure design and probabilistic assessments of the structural response to both fatigue and extreme loads.

Design optimization of structures subject to probabilistic problem variables and parameters, sometimes called optimization under uncertainty, is in general a large field of research at the intersection of two larger fields, optimization and probabilistic design. One main distinction is between *robust* design optimization (RBO) (Ben-Tal et al., 2009; Zang et al., 2005) and *reliability-based* design optimization (RBDO) (Choi et al., 2007; Valdebenito and Schuëller, 2010a). We will generally restrict our discussion to RBDO and reliability methods, but refer to studies on RBO and robust methods where necessary or appro-

priate. Furthermore, given the extensive research on more general applications of reliability analysis, optimization and RBDO (or optimization under uncertainty more generally), we will focus on previous studies concerning wind turbines. For a more expansive overview of structural reliability and RBDO applied to wind turbines than the one following below, the interested reader is referred to Jiang et al. (2017); Leimeister and Kolios (2018); Hübler (2019) and Hu (2018).

A substantial amount of the literature for both structural reliability analyses and RBDO of offshore wind turbines (OWTs) has

focused on aspects other than support structure design. Areas such as, e.g., blade design (Ronold et al., 1999; Toft and Sørensen, 2011; Dimitrov, 2013; Hu et al., 2016; Caboni et al., 2018), foundation design (Yoon et al., 2014; Carswell et al., 2015; Depina et al., 2016; Haja et al., 2019; Depina et al., 2017; Velarde et al., 2019), component design (Kostandyan and Sørensen, 2011; Rafsanjani et al., 2017; Lee et al., 2014; Li et al., 2017), system/wind farm aspects (Sørensen et al., 2008), inspection and maintenance planning and probabilistic tuning/optimization of safety factors (Sørensen and Tarp-Johansen, 2005; Márquez-

Domínguez and Sørensen, 2012; Veldkamp, 2008) have all been studied. As for support structure design specifically, though most structural analyses of OWTs remain deterministic, there has been a number of studies incorporating reliability-based (or otherwise probabilistic) approaches. For the most part, the reliability-based analyses can be divided into two categories. Firstly, there are studies using simplified probabilistic models where the uncertainty in the response is assumed to be a product of the underlying stochastic variables and the deterministic response variable (e.g. Thöns et al. (2010); Sørensen and Toft (2010);

Wandji et al. (2016); Yeter et al. (2016) as well as several of the previously cited studies). Note that the basis for this kind of factorization can be justified partially or entirely depending on both the type of response variable and the type of stochastic variable. For example, in the case of Yeter et al. (2016) the stochastic variables are mostly either modeling or simulation errors or directly originating within the analytical expressions for the response variable. Hence the level of approximation involved in this kind of probabilistic modeling varies. While not strictly in the same category, studies where the fatigue calculation is based

on crack propagation models and an assumption that the stress cycles follow a Weibull distribution, allowing exact limit state



expressions to be derived, should be mentioned (e.g. Dong et al. (2012)). Generally, these simplifications are done in order to be able to solve the reliability problem using first/second order reliability methods (FORM/SORM) in a computationally feasible way. In the second category of studies, the response itself is simplified while generally no particular assumptions about the stochastic nature of the response are made. This has been done through static response modeling (e.g. Wei et al. (2014);

Kim and Lee (2015)), but usually the response is replaced by surrogate models of some kind (e.g. Kolios (2010); Teixeira et al. (2017); Morató et al. (2019)). The use of surrogate models is often done in order to be able to solve the reliability problem by sampling methods, generally requiring a large number of response evaluations, but surrogate models also make FORM/SORM more computationally practical. Note that this division of reliability methodology is not strict, Thöns et al. (2010) also makes use of surrogate modeling for example, nor does it cover all approaches, but it is useful as an indicator

for one of the fundamental struggles that all the aforementioned studies have reckoned with: The fidelity of the probabilistic modeling versus the fidelity of the underlying structural analysis.

Only a limited number of studies applying RBDO to OWT support structure design have been made. In a series of studies, Yang and collaborators investigated optimization of a tripod support structure with probabilistic constraints. In Yang et al. (2015), RBDO was performed and in Yang and Zhu (2015), RBO was performed. In both cases a Gaussian process (Kriging)

surrogate model was used for the response and Monte Carlo sampling was used for the reliability calculation. In Yang et al. (2018), RBDO was once again performed with a Gaussian process surrogate model, but in this case the reliability calculation was done using a Fractional Moment method in order to reduce the number of system evaluations required. All three studies used the heuristic optimization method called Multi-island genetic algorithm and the reliability calculations were done for each step in the optimization loop, creating a nested two-loop structure. As one might expect from a heuristic method, the number

of iterations required to solve even the deterministic optimization problem is rather large given the small number of design variables used and this is much more pronounced in the case of the stochastic optimization. This means that the method is rather computationally inefficient, especially considering the number of system evaluations required by the reliability calculation and the genetic algorithm at each iteration. However, due to the use of the surrogate model, this practical issue is overcome, if still apparent. Though not an application to OWTs, it is also worth mentioning the study of RBDO applied to offshore monopod

towers in Karadeniz et al. (2010b) and applied to jacket structures in Karadeniz et al. (2010a). Here the limit state functions are formulated analytically and a nested two-loop approach using gradient-based optimization (in this case sequential quadratic programming, SQP) and FORM is used.

The previous work on RBDO for OWTs (for both support structures and otherwise) demonstrate that these methods can obtain optimal designs that are both more robust/safe with respect to uncertainties than designs optimized under deterministic

criteria and more tailored to specific design conditions than deterministic designs using safety factors. However, so far (and this is particularly true for support structure designs) no studies have taken advantage of recent advances in deterministic structural optimization methodology. In particular, some recent studies have demonstrated the viability and, in most cases, advantages of analytical sensitivities in gradient-based formulations. This has been shown for static (Sandal et al., 2018), quasi-static (Oest et al., 2017) and dynamic (Chew et al., 2016) loading conditions (see also Oest et al. (2018) for a comparison

of these three approaches). In general, these approaches make the design optimization problem more efficient and stable,





though because of the added conceptual complications these methods have yet to be applied in studies considering a more realistic and comprehensive set of loading conditions. A study founded on gradient-based optimization that *does* consider a more comprehensive set of loading conditions, but does not utilize analytical sensitivities, was performed by Häfele et al. (2019). They used a Gaussian process surrogate model to simplify the response, thus making the analysis computationally

feasible. This study also used a more complicated and (arguably) more realistic objective function, modeling the cost of the support structure in a more detailed way than the strictly steel mass-/volume-based approaches that are otherwise commonly used. However, it was seen that, at least with the particular cost formulations used, the solution was more or less the same as when a simpler mass-based objective function was used. Another recent study regarding deterministic support structure optimization was done in Couceiro et al. (2019). Like the previous study, completely analytical sensitivities were not used. The

plausibility of more comprehensive code checks for design optimization under dynamic loading was in this case demonstrated by a simplified fatigue extrapolation procedure and an aggregation of time-dependent stress constraints (for ultimate limit state analysis) into a single constraint per stress time series. All these studies have been focused on bottom-fixed structures (jackets in particular, though the methodologies are easily transferable to monopiles) and it is unclear what level of adaptation is necessary to extend these formulations to floating structures.

As seen in several of the cited studies above, the use of surrogate modeling to simplify the response analysis has become more common recently. For optimization and reliability analysis, and all the more so for RBDO, this is a natural way to make the problems more computationally tractable when faced with having to perform a large number of time-consuming simulations. However, surrogate modeling is increasingly also proposed for basic structural analysis due to the large number of environmental states that need to be checked for certification according to design standards (e.g. International Electrotechnical

Commission (2009)). For example, Toft et al. (2016a) used a response surface based on Taylor expansions and Gaussian process regression was used by Huchet et al. (2019) and Teixeira et al. (2019) for fatigue design and by Abdallah et al. (2019) for ultimate limit state (ULS) design. Though there are some challenges regarding the number of samples required to build an accurate model, this can be alleviated by efficient design of experiment and/or adaptive methods. The overall indication seems to be that surrogate modeling, and particularly Gaussian process regression, provides a viable strategy for simplifying

the structural analysis in design problems.

In summary, while considerable work has gone into improving the various analyses and methods involved in RBDO for support structures, there is a very limited amount of studies that connect these pieces together. In particular, the work on analytical design sensitivities has not been implemented into RBDO, nor has it been combined with surrogate modeling approaches that make more comprehensive structural analysis and/or reliability analysis computationally feasible. These gaps are what we

intend to explore in the present study. By a very particular formulation of the probabilistic constraints (limit state functions) used for the support structure design optimization, we demonstrate how these constraints can remain analytically differentiable with respect to the design variables while at the same time using a surrogate model for the stochastic variation of the response. By doing so, we retain the advantages of the state-of-the-art deterministic optimization formulations while ensuring that the uncertainties are propagated through the system in a way that makes less simplifications than the commonly used factorization

approaches and without incurring substantial additional computational effort. By assuming that some kind of factorization of



the response is valid locally in design space, a standard double loop RBDO formulation can be applied together with a design-independent Gaussian process surrogate model that makes the inner loop used to solve the reliability problem computationally insignificant. Retraining the surrogate model and repeating the optimization a few additional times then leads to convergence and an optimal design that is feasible with respect to uncertainties in both the loads and the structural modeling. In addition to incorporating more advanced optimization methods to the RBDO problem than has been done previously for OWT support structure design, the current approach can also be seen as a natural middle ground between, on the one hand, the simplified analytical limit state formulations and, on the other hand, the completely surrogate model based limit state formulations; the two most commonly used approaches in reliability analysis and RBDO for OWTs previously.

## 2   Methodology

In the following, we present the basic framework of (deterministic) design optimization for OWT support structures. Then, some aspects of RBDO and surrogate modeling are explained. Finally, the synthesis of these aspects resulting in the proposed RBDO methodology is motivated and presented.

### 2.1   Design optimization of offshore wind turbine support structures

For the task of finding the minimum structural mass $f_{\mathrm{mass}}$ of a topologically fixed design consisting of $N$ circular cross-sections, the following optimization problem can be formulated:

$$\min_{\mathbf{x}} f_{\mathrm{mass}}(\mathbf{x}) \quad \text{such that:}$$

$$A_{\mathrm{lin}}\mathbf{x} \leq \mathbf{b}$$

$$\mathbf{x} \leq \mathbf{x}_u$$

$$\mathbf{x} \geq \mathbf{x}_l$$

$$c_j(\mathbf{x}) \leq 0 \quad \forall j \in \mathcal{J} \tag{1}$$

Here $\mathbf{x}$ are the design variables, $A_{\mathrm{lin}}$ and $\mathbf{b}$ give rise to a system of linear inequality constraints, $\mathbf{x}_u$ and $\mathbf{x}_l$ are upper and lower bounds respectively and $c_j$ are non-linear constraint functions indexed according to some set $\mathcal{J}$. The design variables for this problem will be the diameters $D_i$ and thicknesses $t_i$ of each cross section $i \in \{1, ..., N\}$. The total mass of all $N$ cross sections is calculated as:

$$f_{\mathrm{mass}}(\mathbf{x} = (\mathbf{D}; \mathbf{t})) = \pi\rho \sum_{i=1}^{N} L_i(D_i t_i - t_i^2) \tag{2}$$

where $L_i$ are the (constant) lengths of each structural element with cross-sections given by $D_i$ and $t_i$ and $\rho$ is the material density (assuming a uniform density throughout the structure). Examples of the type of linear constraints that can be represented by $A_{\mathrm{lin}}\mathbf{x} \leq \mathbf{b}$ are limits on the ratio of each $D_i$ to each $t_i$ (the D-t-ratio). The non-linear constraints $c_j$ typically correspond to





safety criteria for ULS and the fatigue limit state (FLS), but often also include constraints on the first eigenfrequency of the structure.

The optimization problem in Eq. (1) can be solved either by gradient-based or gradient-free (heuristic) methods. The former is often preferred when possible, because it leads to faster convergence. Examples of common heuristic methods are genetic

algorithms, particle swarm algorithms and random search. However, we will focus on gradient-based methods going forward. All gradient-based methods require, as the name suggests, the calculation of the gradients of the problem. In a constrained problem like Eq. (1), that means estimating the gradients of both the objective function $f_{\text{mass}}$ and all the constraints. For an objective function like the one stated in Eq. (2) and for any linear constraints, this is a trivial problem. For non-linear constraints, the calculation of gradients (often called sensitivities in the optimization field) can be very difficult, especially when the value of

these constraints depend on output from simulations, as is generally the case for support structure optimization. This difficulty can in principle be accommodated by the use of finite difference methods, where the function values around the current design point are used to get an estimate of the gradient. However, the use of finite difference methods can lead to inaccurate solutions, failure to converge or will at least often require a larger number of function evaluations (computationally costly when simulations are needed for each such evaluation) to obtain the same solution as one would using the exact gradients (see

e.g. Chew et al. (2016)). Additionally, the accuracy of finite difference estimates depend strongly on the chosen step size, the optimal value of which again depends strongly on the (possibly local) properties of the function in question (see e.g. Press et al. (2007) for a general discussion and Oest et al. (2017) for a demonstration of this effect for support structure design). Hence, it is desirable to use analytical sensitivities whenever possible.

It is a well known result (see e.g. Kang et al. (2006)) that when the displacements $\mathbf{u}(t)$ of the structural system under dynamic

loading $\mathbf{S}(t)$ are found by time-integration of the equation of motion, given as

$$M(\mathbf{x})\ddot{\mathbf{u}}(t) + C(\mathbf{x})\dot{\mathbf{u}}(t) + K(\mathbf{x})\mathbf{u}(t) = \mathbf{S}(t) \tag{3}$$

for mass matrix $M$, damping matrix $C$ and stiffness matrix $K$, then the sensitivties of the displacements can be found by time-integration of the following equation:

$$M(\mathbf{x})\frac{\mathrm{d}\ddot{\mathbf{u}}(t)}{\mathrm{d}\mathbf{x}} + C(\mathbf{x})\frac{\mathrm{d}\dot{\mathbf{u}}(t)}{\mathrm{d}\mathbf{x}} + K(\mathbf{x})\frac{\mathrm{d}\mathbf{u}(t)}{\mathrm{d}\mathbf{x}} = \frac{\mathrm{d}\mathbf{S}(t)}{\mathrm{d}\mathbf{x}} - \left(\frac{\mathrm{d}M(\mathbf{x})}{\mathrm{d}\mathbf{x}}\ddot{\mathbf{u}}(t) + \frac{\mathrm{d}C(\mathbf{x})}{\mathrm{d}\mathbf{x}}\dot{\mathbf{u}}(t) + \frac{\mathrm{d}K(\mathbf{x})}{\mathrm{d}\mathbf{x}}\mathbf{u}(t)\right) \tag{4}$$

Hence, if the non-linear constraints can be expressed as analytical functions of the displacements, the sensitivities are obtainable via (possibly repeated) application of the chain rule and finally the solution of Eq. (4). It is presently assumed that the system matrices $M$, $C$ and $K$ are known analytical functions of the design variables, as is the case when the structural analysis is based on finite element modeling with beam elements defined according to Euler or Timoschenko beam theory. If this is not the case, the use of semi-analytical methods (where the gradients of the system matrices are estimated with finite differences) must be

used. For OWT support structures, it was shown in Chew et al. (2016) how the sensitivities of both ULS and FLS constraints could be obtained using the analytical approach described above.



## 2.2 RBDO

The main distinguishing feature, with respect to the problem structure defined in Eq. (1), of optimization under uncertainty, is the addition of a new set of stochastic variables $\theta$ that in general can enter both the objective function and the constraints. In fact, some or all of the design variables in $\mathbf{x}$ could be replaced (or depend on) variables in $\theta$. However, in our case we shall

restrict the discussion to cases where all the design variables are deterministic. It follows that the only $\theta$-dependence must then be in the so far to be determined non-linear constraints $c_j$. In RBDO, the main idea is that we seek to constrain (and/or, in some formulations, optimize) the reliability of the system. The reliability of a structural system is a probabilistic measure of its ability to resist loads. In the most straightforward mathematical representation, this is expressed as the extent to which the load effect $Q$ (usually depending on the response) does not exceed the resistance $R$ (usually depending on the capacity or structural

strength). In a probabilistic setting, this is quantified by the probability of failure $P_f$, defined as:

$$P_f = \text{Prob}(Q - R > 0) \tag{5}$$

Formally, the reliability is the probability of non-failure, $1 - P_f$, though commonly one tends to use $P_f$ rather than the actual reliability in analysis and calculations. Furthermore, since the analogy of load effect and resistance is not always applicable, the notion of failure is usually represented by a limit state function $g$, encoding failure as positive function values, with the

probability of failure as:

$$P_f = \text{Prob}(g > 0) \tag{6}$$

In general, $g$ is a function of both $\mathbf{x}$ and $\theta$ and to calculate $P_f$ requires knowledge of the joint probability distribution $h_\theta$ of all the stochastic variables in $\theta$. An exact estimate of $P_f$ is then given by the integral of $h_\theta$ over the part of its domain where $g > 0$, i.e.:

$$P_f = \int_{g(\mathbf{x},\theta)>0} h_\theta(\theta')\mathrm{d}\theta' \tag{7}$$

The general RBDO problem may then be formalized as:

$$\min_{\mathbf{x}} f_{\text{mass}}(\mathbf{x}) \quad \text{such that:}$$

$$A_{\text{lin}}\mathbf{x} \leq \mathbf{b}$$

$$\mathbf{x} \leq \mathbf{x}_u$$

$$\mathbf{x} \geq \mathbf{x}_l$$

$$c_j(\mathbf{x}) \leq 0 \quad \forall j \in \mathcal{J}_{\text{det}}$$

$$P_{f,j}(\mathbf{x}) \leq P_{f,j}^{\text{max}} \quad \forall j \in \mathcal{J}_{\text{prob}} \tag{8}$$

where $\mathcal{J}_{\text{det}}$ and $\mathcal{J}_{\text{prob}}$ represent the indices of deterministic and probabilistic constraints respectively and $P_{f,j}^{\text{max}}$ are the desired upper bounds on the probabilities of failure $P_{f,j}$.





However, for any but the most trivial limit state functions, the determination of the values of $\mathbf{x}$ and $\theta$ giving $g > 0$, and hence the determination of the integral in Eq. (7), cannot be done exactly. The most straightforward and robust way to accommodate this is through the use of sampling methods. In particular, the family of Monte Carlo and Quasi-Monte Carlo methods are typically used. These methods generally have the property that for a large enough sample size, the resulting estimate $\hat{P}_f$

tends towards the exact value of $P_f$. Unfortunately, *large enough* can be an intractable requirement. While the use of variance reduction techniques can speed up the convergence, as the dimensionality and complexity of the problem grows, the number of samples does too. This can be particularly problematic when one or more simulations are required for each sample. Furthermore, sampling methods do not naturally lend themselves well to gradient-based optimization due to the additional effort involved in the calculation of the gradients of a quantity estimated by sampling. In some cases, when the design variables

are stochastic, the use of what is called score functions for the estimation of design sensitivity is possible, in which case no additional samples are needed (see e.g. Hu (2018)). At the very least, no analytical gradients can be obtained. Hence, it is common to make use of first and second order approximations of the limit state function, making integration over the $g > 0$ region feasible.

### 2.2.1 FORM

The objective of FORM is to approximate the non-linear failure surface, the set of points such that $g > 0$, by a linear function of independent standard normal variables $\mathbf{v}$, derived from the original set of stochastic variables $\theta$. Historically there have been several versions of FORM and related methods (see e.g. Choi et al. (2007) and Enevoldsen and Sørensen (1994), where also more details about FORM in general can be found), but we shall restrict the discussion to the one most commonly used. The main idea is as follows: Construct the set of independent standard normal variables $\mathbf{v} = \{v_i\}$ by applying the transformations

$$v_i = \Phi^{-1}(H_{\theta_i}(\theta_i)) \quad \forall i \in \mathcal{I} \tag{9}$$

where $\Phi^{-1}$ is the inverse of the standard normal cumulative distribution function (CDF), $H_{\theta_i}$ is the CDF of $\theta_i$ and $\mathcal{I}$ is the set of all the indices for the stochastic variables in $\theta$. This particular transformation assumes that the variables in $\theta$ are independent, which is not always the case. For non-independent stochastic variables, a slightly more involved transformation (e.g. the Rosenblatt transformation (Rosenblatt, 1952)) must be used. By substituting $\mathbf{v}$ for $\theta$ in $g$, we obtain $g(\mathbf{x}, \mathbf{v})$. We want

to linearize this function at the point on the boundary between failure and non-failure, $g = 0$, that is closest to the origin in standard normal space, the most probable point (MPP) on the failure surface. This can be found by solving the following optimization problem:

$$\min_{\mathbf{v}} \sqrt{\sum_i v_i^2} \quad \text{such that}$$

$$g(\mathbf{x}, \mathbf{v}) = 0 \tag{10}$$

We denote the optimal point solving the above $\mathbf{v}^*$, and the corresponding minimal distance to the origin $\beta = \sqrt{\sum_i (v_i^*)^2}$ is called the reliability index. The probability of failure is then estimated as $P_f = \Phi(-\beta)$. Some care must be taken in the



application of FORM methods, since this representation is only exact in the case that $g$ is a linear function. For a non-linear $g$, FORM is an approximation, but it is often good enough for many engineering applications. Beyond merely offering a tractable solution to Eq. (7), there are several properties that make FORM desirable for RBDO. Consider for example the behavior of the probabilistic constraints in Eq. (8). $P_f$ will tend to vary over many orders of magnitude, which can be detrimental to

the behavior of many algorithms for gradient-based optimization. The introduction of the reliability index means that we can replace the constraints involving $P_f$ with equivalent ones involving $\beta$, i.e.

$$\beta_j \leq \beta_j^{max} = -\Phi^{-1}(P_{f,j}^{\max}) \tag{11}$$

This substitution has a further advantage when calculating sensitivities. Even without an explicit expression for the derivative of $P_f$ with respect to $\mathbf{x}$, we can make the following observation: In the two cases where the design $\mathbf{x}$ is such that the region $g > 0$

is either very small (very safe designs) or very large (very unsafe designs), then the change in $P_f$ due to a small change in $\mathbf{x}$ is virtually zero. Hence, in these design configurations the sensitivity vanishes, which has a detrimental effect on the optimization since most algorithms will struggle to find new candidate points that lead to measurable changes in the constraints. Using $\beta$ as the constraint function instead gives the following, generally non-vanishing, expression (Enevoldsen and Sørensen, 1994):

$$\frac{\mathrm{d}\beta}{\mathrm{d}\mathbf{x}} = \frac{1}{\left\|\frac{\mathrm{d}g}{\mathrm{d}\mathbf{v}}\right\|} \frac{\mathrm{d}g}{\mathrm{d}\mathbf{x}} \tag{12}$$

However, one problem with the definition of FORM given in Eq. (10), typically called the reliability index approach (RIA), is that it is not always possible to find a configuration $\mathbf{v}$ such that $g = 0$ (within a sufficiently small tolerance). This can lead to slower convergence of the RBDO problem or in the worst cases a lack of convergence at all. To resolve this issue, it is possible to formulate an inverse problem where instead of calculating the reliability index for a given design, one finds the configuration of $\mathbf{v}$ giving the smallest exceedance of $g > 0$ for a given (fixed) reliability index. This is called the performance

measure approach (PMA) and will be explicated below.

### 2.2.2 PMA

The main idea of PMA is to reverse the role of objective and constraint in Eq. (10). If we demand that $\sqrt{\sum_i v_i^2} = \beta^{\max}$ as a constraint, we can instead find the largest possible value of $g$ for which that constraint is satisfied. In other words:

$$\max_{\mathbf{v}} \quad g(\mathbf{x}, \mathbf{v}) \quad \text{such that}$$

$$\sqrt{\sum_i v_i^2} = \beta^{\max} \tag{13}$$

If we again call the solution point $\mathbf{v}^*$ and term the corresponding value of $g$ as $g^*$, then under the assumptions of the validity of FORM it follows that $\mathrm{Prob}(g > g^*) = P_f^{\max}$. Hence, by further demanding that $g^* \leq 0$, we can guarantee that $P_f \leq P_f^{\max}$. The optimization problem in Eq. (13) always has a solution. Aside from the robustness provided by this, PMA has a few other advantages. For example, it can be shown that (see e.g. Frangopol and Maute (2005)):

$$\frac{\mathrm{d}g(\mathbf{x}, \mathbf{v}^*)}{\mathrm{d}\mathbf{x}} = \frac{\partial g(\mathbf{x}, \mathbf{v}^*)}{\partial \mathbf{x}} \tag{14}$$





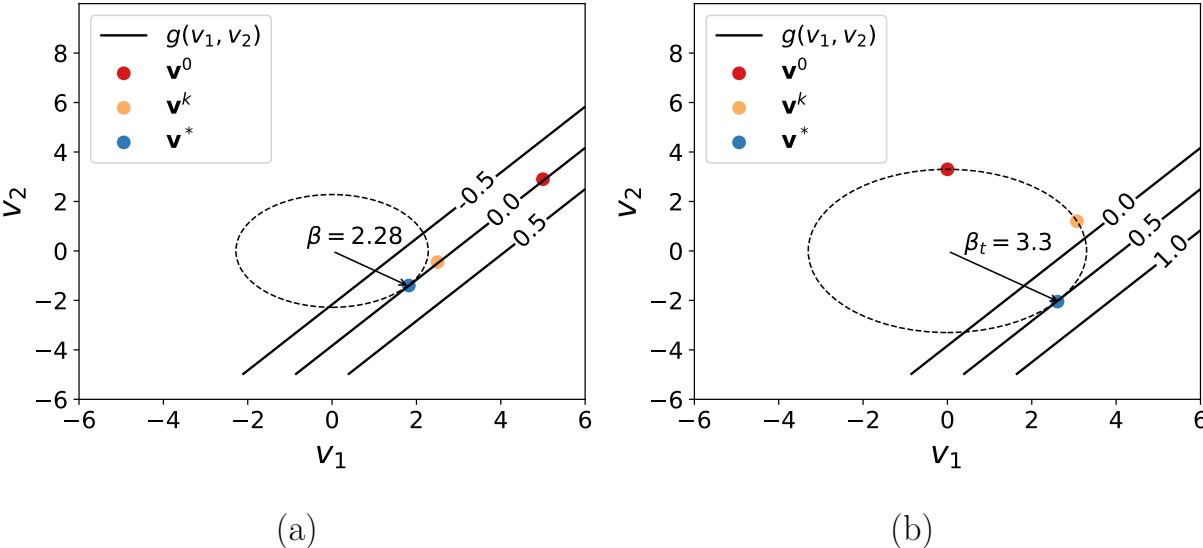

**Figure 1.** The difference between the solutions provided by RIA (a) and PMA (b) for a linear limit state function $g$ with two variables, $(v_1, v_2) = \mathbf{v}$, in standard normal space. The target reliability index for PMA ($\beta_t = 3.3$) is here higher than the RIA solution ($\beta = 2.28$), so the PMA solution finds $g^* > 0$. Also indicated are examples of points visited during the respective optimizations (initial, $\mathbf{v}^0$, intermediate, $\mathbf{v}^k$, and solution points, $\mathbf{v}^*$; different for the two methods), where the displayed points before the solution are feasible, but not optimal.

which simplifies the sensitivity analysis. More generally, for applications to RBDO, PMA tends to perform better (Tu et al. (1999), Youn and Choi (2003) and Lee et al. (2002)). An illustration of the difference between RIA and PMA is made in Fig. 1.

The RBDO problem using PMA can be stated as:

$$\min_{\mathbf{x}} f_{\text{mass}}(\mathbf{x}) \quad \text{such that:}$$

$$A_{\text{lin}}\mathbf{x} \leq \mathbf{b}$$

$$\mathbf{x} \leq \mathbf{x}_u$$

$$\mathbf{x} \geq \mathbf{x}_l$$

$$c_j(\mathbf{x}) \leq 0 \quad \forall j \in \mathcal{J}_{\text{det}}$$

$$g_j(\mathbf{x}, \mathbf{v}^*) \leq 0 \quad \forall j \in \mathcal{J}_{\text{prob}} \tag{15}$$

Where each $g_j$ solves Eq. (13) with $\beta^{\text{max}} = -\Phi^{-1}(P_{f,j}^{\text{max}})$. One potential downside of PMA is that it does not provide a direct estimate of the probability of failure. This is fine for optimization, where being below the threshold is sufficient and where at least one constraint should be at the boundary where $g^* = 0$ for the final solution. However, if one wishes to compare the probability of failure of such an optimized design with the corresponding initial design or a design optimized by deterministic methods, then PMA does not immediately provide a quantitative answer. It only provides a qualitative assessment of whether





or not the probability of failure is above or below the given threshold. To get a quantitative assessment in these cases, we can exploit the approximate linearity of $g^*$, especially close to $g^* = 0$. If we have solved the PMA problem in Eq. (13) once for a target reliability $\beta_t$, obtaining the solution $g^*$, we can then use the secant method to construct an estimate of $\beta_0 \equiv \beta(g^* = 0)$ as:

$$\beta_0 = \beta_t - \frac{\beta_t - \beta'}{g^* - g'^*} g^* \tag{16}$$

where $g'^*$ is the solution of a PMA problem for a target reliability $\beta' \in (\beta_t, \beta_0)$. If the initial $g^*$ is sufficiently small (close to 0) and/or sufficiently linear, then the above will provide a good estimate $\beta_0$ and hence an estimate of the probability of failure as $P_f = \Phi(-\beta_0)$. Otherwise, this procedure can be iterated (setting $\beta_t = \beta'$ and $\beta' = \beta_0$). In such cases, unless the initial $g^*$ is very far away from zero and/or $g$ is highly non-linear, only a few more iterations (1-3) should suffice to get at least 2 digits of

accuracy for $P_f$.

### 2.2.3 Two-loop RBDO vs single-loop RBDO

As is evident from Eq. (15), the current formulation of the RBDO problem consists of two nested loops. One outer optimization problem that solves the design optimization problem under the given constraints and one inner optimization that solves the (PMA) reliability problem to obtain the probabilistic constraints for each iteration. This can be computationally demanding,

even when the convergence of the PMA sub-problem is accelerated by the use of improved optimization methods like the hybrid mean-value algorithm (Youn et al., 2003). For this reason, several alternative solution strategies for RBDO have been proposed (Valdebenito and Schuëller, 2010b). This usually involves either decoupling the two loops into a sequence of deterministic optimization and reliability analysis, most prominently in the sequential optimization and reliability analysis (SORA) method (Du and Chen, 2004), or the use of reformulated single loop approaches, most prominently in the aptly named single loop

approach (SLA) (Chen et al., 1997). All these methods involve some kind of approximation of the FORM-based constraint. While speeding up the convergence significantly compared to conventional two-loop strategies, this can also lead to lack of convergence for some problems (Aoues and Chateauneuf, 2010). On the other hand, SORA seems to be fairly robust, owing in large to the fact that its reliability-based constraint is locally equivalent to the two-loop approach, meaning that as the changes in the design become small from one round of deterministic optimization to the next, the error in the approximation when using

a fixed reliability estimate during the design optimization tends to zero.

### 2.3 Surrogate modeling

Surrogate modeling is generally a vast topic and the interested reader is referred to Wang and Shan (2006) and Marsland (2015) for more general overviews, as well as to Tunga and Demiralp (2005) for the high dimensional model representation approach and Rasmussen and Williams (2006) and Santner et al. (2018) for more detailed looks at Gaussian process regression (GPR).

For applications to RBDO in general Dubourg (2011) and Jin et al. (2003) are instructive.

Focusing our attention to wind turbine applications, it has been common for quite some time to use surrogate modeling due to the computationally demanding simulations required for time domain analysis. Especially for reliability analysis, optimization





and RBDO where the computational effort increases drastically. The most commonly applied types of surrogate models in wind energy have been response surface models (typically second order polynomials), Taylor expansions and (especially more recently) GPR. GPR has many advantages, including the ability to capture non-linearities with higher fidelity and providing an estimate of its own uncertainty by default, but generally requires a larger number of samples to gain a significant advantage

over response surface methods (Kaymaz, 2005). We note that GPR is often referred to as Kriging in the engineering literature. Although for most practical purposes, the two terms can be used interchangeably, GPR is more general. Hence, to avoid specificity where it is not needed, we will use the term GPR.

### 2.3.1   GPR

The essentials of GPR are quite similar to conventional regression methods. We wish to construct a model $y(x)$ for the response

$y$ to some input $x$. However, instead of considering for example a multi-linear or polynomial model plus a simple noise term, one instead considers a more general expansion of the input in some basis $B$ (which could be constant, linear, polynomial or otherwise) plus a realization of a zero-mean Gaussian process $GP$:

$$y = \gamma B(x) + GP(x) \tag{17}$$

where $\gamma$ is a set of basis coefficients. The Gaussian process is determined by its covariance function, which is the product of the

noise parameter $\sigma$ and a kernel function. The kernel function gives the covariance function its main structure by determining the correlation between points $(x, x')$. Usually, these kernel functions are exponentially decaying with the Euclidean distance between the points. In addition to $\sigma$, the covariance function is parameterized by one or more hyperparameters. All in all GPR consists of fitting $\gamma$, $\sigma$ and all the kernel parameters based on a set of training data $\{y_i, x_i\}$, where in general each input $x_i$ can be multi-dimensional. These parameters are fit using maximum likelihood estimation, though finding optimal parameters often

requires the use of global optimization methods in order to fully consider the range of possible parameter values. The fitted covariance function of the GPR model, in particular the value of $\sigma$, provides a natural estimate of the inherent uncertainty (or expected error) of the surrogate model, which can then be used to establish confidence/prediction intervals for predicted model responses to new inputs. An illustration of GPR is given in Fig. 2.

### 2.3.2   Design of experiment

As noted previously, GPR can require a large number of samples to attain its desired fidelity. For this reason, it is common to apply specialized sampling techniques, together usually referred to as design of experiment (DOE), that sample the input space more efficiently and thereby require less samples than, e.g., uniform random sampling. Depending on the desired outcome, one could for instance opt for importance sampling (most useful in this case if it is known that only a certain region of the input space is of interest, e.g. for a reliability analysis where one mainly wishes to use the surrogate model around the failure

surface) or a space-filling approach like Latin hypercube sampling or Quasi-Monte Carlo sampling (these are most useful when as wide coverage of the input space as possible is needed, e.g. for optimization where the region of interest is likely to shift dynamically). A comparison between Latin hypercube sampling and Quasi-Monte Carlo sampling was performed in





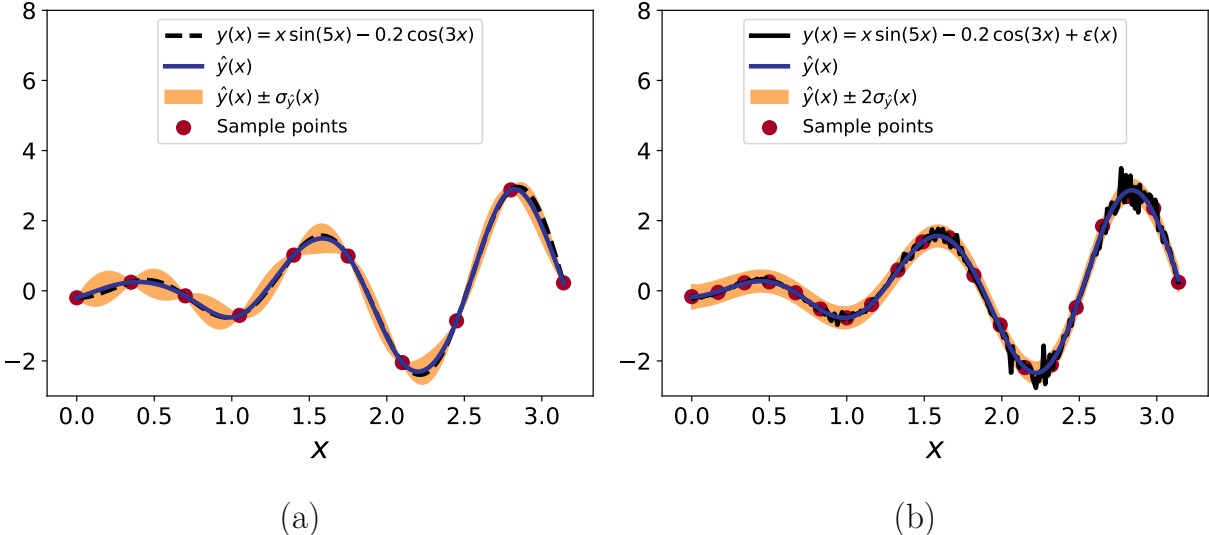

(a)                                                                                   (b)

**Figure 2.** GPR demonstrated on two related test functions, one with no noise (a) and one with noise (b). In the former case, 10 sample points is enough for a very good estimate (the real function, $y$, being within one standard deviation of the estimate, $\hat{y}$, throughout). Note also how the uncertainty descreases around the sample points in this case, due to the lack of noise. In the second case, more samples are needed for a good estimate. The noisy function does at one point exceed even two standard deviations away from the estimate, but the underlying (non-noisy) function is well estimated. Note also how the uncertainty level, while more or less constant, is not higher than it was away from the sample points of the non-noisy case.

Kucherenko et al. (2015), where it was found that Latin hypercube sampling can give better or more efficient results for certain types of problems, but that Quasi-Monte Carlo sampling was otherwise equal or superior and generally more robust when the problem could not be classified a priori. Latin hypercube sampling has been more common for wind energy applications, but a Quasi-Monte Carlo sampling method based on the Sobol sequence was used in Müller and Cheng (2018).

## 2.4 Proposed RBDO framework

In the following, we will explain the details of our proposed framework for RBDO of OWT support structures. However, we begin with a few remarks that serve to motivate this approach.

### 2.4.1 Motivation

Considering the state-of-the-art for reliability analysis and RBDO for OWTs more generally, we can make a few summary observations based on the previous discussion: Firstly, the vast majority of studies make use of either simplified analytical limit state functions (allowing more easily the use of FORM and making the probabilistic constraints easier to combine with design optimization) or surrogate models that completely replace simulation output (usually combined with sampling-based reliability analysis). Secondly, when not based on heuristic optimization methods (as has been the case for all RBDO studies concerning



the design of support structures specifically), gradient-based design optimization as part of RBDO has not utilized analytical sensitivities. Thirdly, little to no use of PMA for reliability analysis or more advanced RBDO methods like SORA or SLA have been made, despite their notable advantages.

What can be concluded from this? Simply put, considerable progress could be made by making the state-of-the-art for OWT
RBDO, and for support structure design in particular, more in line with the general state-of-the-art. However, this should be done in a way that maintains some of the OWT-specific developments made in previous optimization studies. Furthermore, by combining elements from all these sources, it could be possible to obtain a synthesized methodology that retains many of the individual advantages. However, this requires a new approach because of the ways in which the previous methods seem incompatible: It is, e.g., seemingly not possible to use analytical sensitivities if the simulation output is replaced by surrogate
models.

### 2.4.2 Key simplification

Suppose that all relevant limit state functions $g_j$ can be written in the form $g_j = Q - R$, which is generally the case for support structure design, with $Q$ and $R$ being the load effect and resistance as before. Furthermore, for simplicity (and since this is usually the case), assume that while both $Q$ and $R$ are functions of the design $\mathbf{x}$ and the stochastic variables $\theta$, only $Q$ is
determined by simulations. We then make the following simplification:

$$Q(\mathbf{x}, \theta) = Y(\theta)\tilde{Q}(\mathbf{x}) \tag{18}$$

where $Y$ is some arbitrary unknown function with the property that $Y(\bar{\theta}) = 1$, with $\bar{\theta}$ as the mean values of $\theta$, and $\tilde{Q}(\mathbf{x}) = Q(\mathbf{x}, \bar{\theta})$ is the mean response at the specific design $\mathbf{x}$. A simple example of how such a factorization makes sense locally is shown in Fig. 3. What are the implications of this assumption? Firstly, note that this assumption is consistent with the common
simplified limit state functions where the stochastic response is modeled as the product of the stochastic variables $\theta$ and the design dependent mean response. However, in our case we make no assumption about the functional representation of this factorization. Hence, this should allow for a higher fidelity representation of how stochastic variables input to the system are propagated through the response estimation. Secondly, while this is indeed a simplification which cannot in general be assumed valid, previous studies detailing how the fatigue damage distribution of OWT support structures changes when the design is
modified (Stieng and Muskulus (2018) and Stieng and Muskulus (2019)) indicate that this kind of proportional scaling is a reasonable assumption as long as the design does not change too much. Furthermore, it is not unreasonable to make a similar assumption for extreme loads. Thirdly, this factorization makes it possible to fit a surrogate model of the response to variations in the stochastic variables only, while the design-dependent part of the response remains as in a deterministic setting. On the one hand, this means that we can fix the design and fit our surrogate model as:

$$Y(\theta^s) = \frac{Q(\mathbf{x}, \theta^s)}{\tilde{Q}(\mathbf{x})} \tag{19}$$

where for each sampled point $\theta^s$ we estimate the total response $Q$ and then factor out the design-dependent mean response. This greatly reduces the dimensionality of the surrogate modeling problem, since we do not have to also sample different



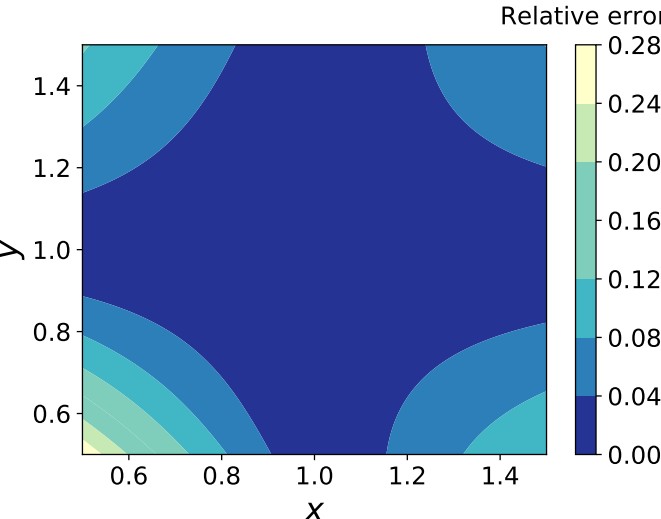

**Figure 3.** An example of factoring out the dependence of one variable from a non-seperable expression by using GPR to fit this unknown factor. The figure shows the relative error when approximating $(x+y)^2$ as $f(y)(x+1)^2$, around $x=1$, with $f(1)=1$ and otherwise unknown. Note the accuracy of this representation around (1,1) and in general the moderate error level as we move away from this region.

values of $\mathbf{x}$. Since the fit is design-independent given the underlying simplifications, we may then say that we obtain a quasi-global (in design-space) surrogate model that can be used throughout a design optimization procedure, greatly reducing the computational effort of any reliability calculation. On the other hand, the separation of stochastic and deterministic response means that for the estimation of design sensitivities we have the property that

$$5 \quad \frac{\partial Q(\mathbf{x},\theta)}{\partial \mathbf{x}} = Y(\theta)\frac{\partial \tilde{Q}(\mathbf{x})}{\partial \mathbf{x}} \tag{20}$$

Hence, the use of analytical design sensitivities becomes possible. Finally, note that while the simplification is expected to lose accuracy as the design moves further and further away from the initial configuration where the surrogate model was fit, the mean response remains exact. This is not the case when a surrogate model fit replaces the simulated response entirely. Hence, for use in RBDO, the factorization in Eq. (18) is going to behave at worst like a deterministic optimization that includes some
10 simplified reliability estimate (based on $Y$) that modifies both the constraint value and the constraint gradients, in a way not too different from SORA and SLA.





### 2.4.3 Formal statement

Our overall proposed framework is based on the previously stated PMA-based RBDO problem in Eq. (15), restated here for convenience:

$$\min_{\mathbf{x}} f_{\text{mass}}(\mathbf{x}) \quad \text{such that:}$$

$$A_{\text{lin}}\mathbf{x} \leq \mathbf{b}$$

$$\mathbf{x} \leq \mathbf{x}_u$$

$$\mathbf{x} \geq \mathbf{x}_l$$

$$c_j(\mathbf{x}) \leq 0 \quad \forall j \in \mathcal{J}_{\text{det}}$$

$$g_j(\mathbf{x}, \mathbf{v}^*) \leq 0 \quad \forall j \in \mathcal{J}_{\text{prob}}$$

$g_j$ is now defined as

$$g_j(\mathbf{x}, \theta) = y_j(\theta_q)q_j(\mathbf{x}) - r_j(\mathbf{x}, \theta_r) \tag{21}$$

with $y_j$ as a surrogate model defined and fit according to Eq. (18) and Eq. (19) and $\theta_q$ and $\theta_r$ are the stochastic parameters in $\theta$ for the load effect and the resistance respectively. Note that we can obtain $\theta_i = H_{\theta_i}^{-1}(\Phi(v_i))$, so that even though the solution of the reliability sub-problem resulting in $\mathbf{v}^*$ is performed in standard normal space, it is never necessary to obtain $y_j$ as a function

of $\mathbf{v}$. To ensure that the RBDO problem is solved with sufficient accuracy, specifically that the final design is actually feasible with respect to the probabilistic constraints, the procedure can be repeated several times; fitting a new surrogate model at the solution of the previous RBDO loop and starting a new RBDO loop from this design point. The overall method is compactly stated as Algorithm 1.

## 3 Testing and implementation details

The design optimization performed in this study will in all cases be based on output from time domain simulations of finite element models. These have been implemented as assembled Timoschenko beam elements with 6 degrees of freedom at each end of each element. The analysis is based on Newmark integration and uses a consistent mass matrix and a Rayleigh damping matrix with mass and stiffness proportionality scaled according to the first two eigenmodes. A typical finite element, including variables, is shown in the left panel of Fig. 4 and a more general representation of the OWT system is shown in the right panel

of Fig. 4.

### 3.1 Models and loads

To test the proposed methodology, two main cases will be used. The first of these cases is a simplified model based on a uniform section of a monopile support structure, initially uniform in its cross-sectional dimensions and with uniform lengths for each





---

**Algorithm 1** RBDO with surrogate modeling and factorized limit state functions

---

Generate $N_s$ samples $\theta^s$ using a specified DOE

Initialize     $\mathbf{x_0}$

Initialize     $f_0 = M(\mathbf{x_0})$

Initialize     $\Delta y$

Initialize     $\Delta f$

$y_{\text{new}}$ = SurrogateModelFit($\theta^s, \mathbf{x_0}$)

**while** $(\Delta f < f_{\text{tol}})$ and $(\Delta y < y_{\text{tol}})$ **do**

    $[\mathbf{x_{\text{sol}}}, f_{\text{sol}} = f(\mathbf{x_{\text{sol}}})] = $ ReliabilityConstrainedOptimization($\mathbf{x_0}, y_{\text{new}}$)

    $y_{\text{old}} = y_{\text{new}}$

    $y_{\text{new}}$ = SurrogateModelFit($\theta^s, \mathbf{x_{\text{sol}}}$)

    $\Delta f = \left| 1 - \frac{f_{\text{sol}}}{f_0} \right|$

    $\Delta y = |y_{\text{new}} - y_{\text{old}}|$

    $\mathbf{x_0} = \mathbf{x_{\text{sol}}}$

    $f_0 = f(\mathbf{x_0})$

**end while**

Where ReliabilityConstrainedOptimization() solves Eq. 15

---

element. This is meant to demonstrate the basic idea of the method without having to consider realistic designs. This model will be referred to below as the **Simple Beam**. The second case is a simplified but reasonably realistic representation of the OC3 Monopile (Jonkman and Musial, 2010) with the cross-sectional dimensions of each segment initially corresponding to the OC3 design, i.e. with a uniform monopile segment and a linearly tapered tower segment. The element lengths are consistent within each major segment, but differ between the tower and monopile. Furthermore, this model also includes a point mass on the top of the tower, with mass and inertia properties meant to represent the NREL 5MW turbine (Jonkman et al., 2009). This model will be referred to as the **OC3 Monopile**. Some of the basic properties of these two models are listed in Table 1 and the material properties are consistent with the ones in Jonkman and Musial (2010). The models are fixed (clamped) at one end (at a location that corresponds to the mudline for the OC3 Monopile), i.e. there is no modeling of soil included. Both models are loaded at the top with force and moment time series extracted from fixed rotor simulations of the NREL 5MW turbine subject to turbulent wind fields within the aero-elastic software FEDEM Windpower (Fedem Technology, 2016). Wave loads are represented by a horizontal force time series, applied at a location corresponding to the bottom of the tower in the OC3 Monopile and at an analog location for the Simple Beam. This force has been tuned to give an equivalent moment at the lower end of the structures as the integrated contribution of all horizontal wave forces along the height of the water column at each instant in time. These forces are calculated from the Morison equation, based on wave kinematics sampled from the JONSWAP spectrum and including Wheeler stretching. The inertia and drag coefficients are 2.0 and 0.8 respectively. The duration of the applied loads are 600 seconds after the removal of initial transients. Up to four different loading scenarios are included in the analysis, with environmental data based on the Ijmuiden Shallow Water Site (Fischer et al., 2010) and using



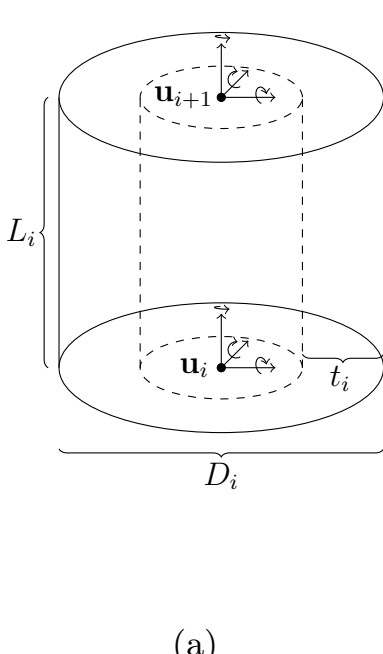

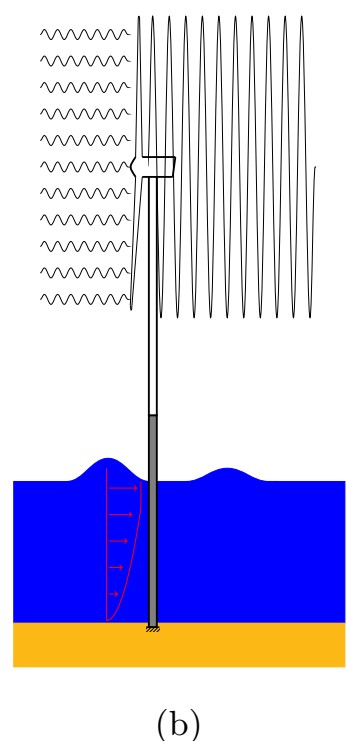

(a)                                    (b)

**Figure 4.** Beam element with design variables $(D_i, t_i)$, length $L_i$, nodal coordinates $\mathbf{u}$ and coordinate systems indicated (a) and the offshore wind turbine system and environment (b).

different random seeds for each realization of wind and wave conditions. The probabilities of occurrence for fatigue estimation have been renormalized so that they sum to 1 for the reduced set of cases. These loading scenarios are summarized in Table 2.

### 3.2   Constraints

As indicated in Eq. (15), the optimization is constrained with upper and lower bounds on the design variables. This is on the one
5    hand done from a theoretical point of view in order that the final designs will be somewhat realistic with respect to practical constraints related to manufacturing, transportation and installation that are not specifically accounted for in the modeling. On a more practical side, this is also done because the finite element simulations can become ill-behaved for certain more extreme variable combinations. The lower bounds are for both models set at 70 % of the smallest initial diameter and thickness respectively. The upper bounds are set at 150 % of the largest diameter and thickness respectively. As for linear constraints, we
10   will in some cases include an upper limit on the D-t-ratio of 120, consistent with the NORSOK standard (Standards Norway, 2013). In terms of non-linear constraints, we consider upper bounds on the accumulated 20-year fatigue damage and on the maximum bending moment. Fatigue calculations are done as follows: From the displacements obtained as the solutions to



Eq. (3), the internal forces and moments $S_{\mathrm{in}}$ are obtained by multiplication with the element stiffness matrices $K^{\mathrm{e}}$ as:

$$S_{\mathrm{in}} = K^{\mathrm{e}} u^{\mathrm{e}} \tag{22}$$

From the components of $S_{\mathrm{in}}$ corresponding to the axial force $S_{\mathrm{ax}}$, in- and out-of-plane moments $S_{\mathrm{ip}}$ and $S_{\mathrm{op}}$, the normal stress $\sigma_n$ is then identified as:

5    $$\sigma_n = \frac{S_{\mathrm{ax}}}{A_c} + \frac{D}{2I_c}(S_{\mathrm{ip}} - S_{\mathrm{op}}) \tag{23}$$

where $A_c = \pi((\frac{D}{2})^2 - (\frac{D}{2} - t)^2)$ is the cross-sectional area and $I_c = \frac{\pi}{4}((\frac{D}{2})^4 - (\frac{D}{2} - t)^4)$ is the second moment of area. Rainflow counting (Rychlik, 1987) is then applied to the normal stress time series, resulting in a set of amplitudes $\Delta\sigma_i$ that correspond

**Table 1.** Properties of the two models used in the study.

| Property | Simple Beam | OC3 Monopile |
|---|---|---|
| Number of elements | 6 | 14 |
| Number of monopile elements | 6 | 3 |
| Number of tower elements | 0 | 11 |
| Lengths of monopile elements [m] | 10 | 10 |
| Lengths of tower elements [m] | N/A | 7.05 |
| Initial diameter of monopile elements [m] | 6.0 | 6.0 |
| Initial thickness of monopile elements [m] | 0.06 | 0.06 |
| Initial diameter of tower bottom [m] | N/A | 6.0 |
| Initial diameter of tower top [m] | N/A | 3.87 |
| Initial thickness of tower bottom [m] | N/A | 0.027 |
| Initial thickness of tower top [m] | N/A | 0.019 |

**Table 2.** Properties of the loading scenarios. IEC DLCs refer to International Electrotechnical Commission (2009).

| Property | Scenario 1 | Scenario 2 | Scenario 3 | Scenario 4 |
|---|---|---|---|---|
| Type of analysis | Fatigue | Fatigue | Fatigue | Extreme load |
| IEC DLC | 1.2 | 1.2 | 1.2 | 1.3 |
| Mean wind speed [m/s] | 4 | 12 | 18 | 18 |
| Turbulence intensity [%] | 20.4 | 14.6 | 13.6 | 20 |
| Significant wave height [m] | 0.97 | 1.57 | 2.56 | 2.56 |
| Peak period [s] | 5.65 | 5.79 | 7.0 | 7.0 |
| Spectral peakedness | 3.3 | 3.3 | 3.3 | 3.3 |
| Surface current speed [m/s] | 0.0 | 0.0 | 0.0 | 0.6 |
| Probability of occurrence | 0.47 | 0.41 | 0.12 | N/A |





to the differences between stresses at particular times (a stress cycle), encoded in the vectors $t_{\text{peak}}$ and $t_{\text{valley}}$, i.e.

$$\Delta\sigma_i = \sigma_n(t_{\text{peak}}(i)) - \sigma_n(t_{\text{valley}}(i)) \tag{24}$$

Unlike what is otherwise common practice, these amplitudes are not binned. This is in order to facilitate the sensitivity analysis. The incurred fatigue damage $F$ is then estimated by use of the Palmgren-Miner linear summation rule, where the contribution

from each stress cycle is given by application of the appropriate SN-curves and thickness correction (Det Norske Veritas, 2016):

$$F = \sum_i \frac{n_i}{a_i} \Delta\sigma_i^{-w_i} \left(\frac{t}{t_{\text{ref}}}\right)^{-kw_i} \tag{25}$$

where $n_i$ is either 0.5 or 1.0 depending on whether the given cycle is a half or full cycle, $a_i$ is a constant, $w_i$ is the Wöhler exponent, $t_{\text{ref}}$ is the reference thickness below which no correction is necessary and $k$ is the thickness correction exponent. The

total lifetime fatigue damage, $F_{\text{tot}}$, from all considered environmental states, $E$, is then:

$$F_{\text{tot}} = T_{\text{lf}} \sum_E F(E) P_{\text{occ}}(E) \tag{26}$$

where $T_{\text{lf}}$ is a factor scaling up from simulation time to 20-year lifetime, $F(E)$ evaluates Eq. (25) for each state $E$ and $P_{\text{occ}}$ are the corresponding probabilities of occurrence for these states. The limit state function for fatigue, measuring the extent to which the lifetime fatigue damage exceeds the fatigue resistance $\Delta_{\text{F}}$, used as a constraint in the deterministic optimization

problem is hence:

$$F_{\text{tot}} - \Delta_{\text{F}} \leq 0 \tag{27}$$

For the maximum bending moment, the calculation is based on $S_{\text{ip}}$ as obtained from Eq. (22). However, since the use of global maxima can cause problems with smoothness (the global maximum may not always change smoothly) and since checking the bending moment at every time step would be very time consuming, a compromise is made. In particular, the Kreisselmeier-

Steinhauser function (Kreisselmeier and Steinhauser, 1979) is used to smoothly aggregate the bending moment time series into an upper envelope of the maximum:

$$M_{\text{KS}} = S_{\text{ip,max}} + \frac{1}{r} \log \left\{ \sum_i \exp\left((S_{\text{ip}}(t_i) - S_{\text{ip,max}})r\right) \right\} \tag{28}$$

where the subscript max denotes the global maximum and $r$ is a constant controlling the accuracy of the approximation. The above expression approaches $S_{\text{ip,max}}$ from above as $r$ approaches infinity. For computations, $r$ is typically taken to be

50-200 depending on desired accuracy (see Couceiro et al. (2019) for a discussion of this for OWT applications). Note that, algebraically speaking, $S_{\text{ip,max}}$ cancels out[1]. Taking Eq. (28) as an estimate of the maximum bending moment, we then compare

---

[1]The nominal expression for the Kreisselmeier-Steinhauser function does not actually include the global maximum as it does here, but for improved numerical performance it has been added (first term) and subtracted (exponential term) as suggested in the original study (Kreisselmeier and Steinhauser, 1979).





this with the NORSOK design criterion for tubular members subject to bending (Standards Norway, 2013). The calculation for the bending resistance uses the one for a D-t-ratio of 120 (for realistic D-t-ratios exceeding this value, the changes are negligible). The resulting limit state is:

$$M_{\text{KS}} - \frac{Z}{\gamma_{\text{M}}}\left(0.94 f_y - 0.76\frac{120}{E}f_y^2\right) \leq 0 \tag{29}$$

where $Z = \frac{1}{6}(D^3 - (D-2t)^3)$ is the plastic section modulus, $\gamma_{\text{M}}$ is the material factor and $f_y$ is the yield strength of the material. The material factor is fixed at 1.45.

### 3.3 Sensitivity

The estimation of gradients for the objective function as given in Eq. (2) and the linear constraints is trivial. For the non-linear constraints, it is mainly a question of repeated application of the chain rule as well as the rule of total derivatives for multivariate functions. See e.g. Chew et al. (2016) for details of how this can be done (the only modification in our case being the additional level added by Eq. (28), which is easily differentiated). However, note that, due to how the displacement sensitivity is calculated in Eq. (4), regardless of location in the structure, there is always a dependence on each design variable for every non-linear constraint. Hence, none of these derivatives are zero in general.

### 3.4 Probabilistic aspects and uncertainty modeling

For the RBDO formulation based on PMA, the limit state functions represented by Eq. (27) and Eq. (29) can be directly used, with the understanding that $F_{\text{tot}}$, $\Delta_{\text{f}}$, $M_{\text{KS}}$ and $f_y$ become probabilistic quantities. Using the notation in Eq. (21), we have:

$$g_{\text{fls}} = y_{\text{fls}}(\theta_q)\tilde{F}_{\text{tot}}(\mathbf{x}) - \Delta_{\text{F}} \tag{30}$$

$$g_{\text{uls}} = y_{\text{uls}}(\theta_q)\tilde{M}_{\text{KS}}(\mathbf{x}) - \frac{Z(\mathbf{x})}{\gamma_{\text{M}}}\left(0.94 f_y - 0.76\frac{120}{E}f_y^2\right) \tag{31}$$

where for simplicity we define $\Delta_{\text{F}} := \theta_{\Delta_{\text{F}}}$ and $f_y := \theta_{f_y}$. The sensitivities then follow from Eq. (14), Eq. (20) and the above discussion. In this study, a target reliability index of $\beta^{\text{max}} = 3.3$, corresponding to $P_f = 4.8 \cdot 10^{-4} (\approx 5 \cdot 10^{-4})$, will be used for the solution of the PMA subproblem in Eq. (13) as part of Algorithm 1. The uncertainties that are included in the response are global stiffness, global damping and turbulence intensity. The uncertainty modeling will be detailed below.

#### 3.4.1 Global stiffness

The uncertainty in stiffness is assumed to come from uncertainty in soil stiffness. Though no soil modeling is included in the present analysis, this mainly introduces a shift of the mean stiffness and hence one may still consider the impact of a stochastic uncertainty in the soil stiffness. The effect of soil pile stiffness on the fundamental eigenfrequency of a monopile was discussed in Kallehave et al. (2015). The expected range of the fundamental frequency was found to be $[0.937, 1.045]$ as a ratio of its mean value. If we symmetrize this as $[0.94, 1.06]$ and use the fact that global stiffness of a monopile is proportional to the square of the fundamental eigenfrequency, we can then obtain the expected range of global stiffness as $[0.88, 1.12]$ as a ratio of




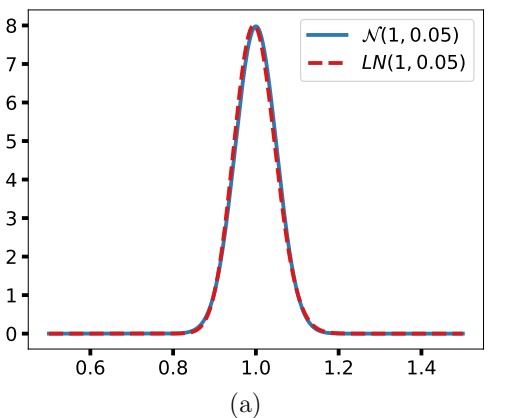
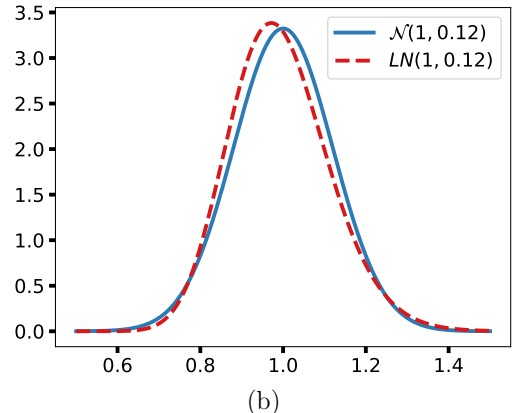

**Figure 5.** Normal distribution vs Lognormal distribution sharing mean and standard deviation: Mean 1.0 and standard deviation 0.05 (a); mean 1.0 and standard deviation 0.12 (b).

its mean value. Taking this to be a 98% confidence interval and, for lack of other information, assuming that the uncertainty in global stiffness follows a Normal distribution, we obtain that $\theta_{\text{stiff}} \sim \mathcal{N}(1, 0.052)$, i.e. a coefficient of variation (CoV) of 0.052. As an independent confirmation, this is close to, if a bit higher, than what would be obtained from Andersen et al. (2012) and Damgaard et al. (2015) (each giving a coefficient of variation of about 0.04).

### 3.4.2   Global damping

For the uncertainty in global damping, the two main contributions are assumed to come from aerodynamic damping and soil damping. Expected ranges of the damping coefficients corresponding to these two sources can be obtained from Chen and Duffour (2018) as $[4.0, 8.0]$ for aerodynamic damping (in the fore-aft direction for operational conditions) and $[0.17, 1.30]$ for soil damping, both given as percentages of critical damping. Assuming, as above, that these ranges correspond to 98% confidence

intervals and that the uncertainty can be modeled (for lack of better knowledge) as following a Normal distribution, then we obtain $\theta_{\text{damp,aero}} \sim \mathcal{N}(6, 0.86)$ and $\theta_{\text{damp,soil}} \sim \mathcal{N}(0.735, 0.243)$. Summing these contributions and adding also a constant (deterministic) structural damping of 1.0, the final result is $\theta_{\text{damp}} \sim \mathcal{N}(7.735, 0.89)$, a CoV of 0.115. The soil uncertainty obtained here is about the same as would be derived from Damgaard et al. (2015). The aerodynamic damping is harder to verify with additional sources and in principle the level of uncertainty is expected to be wind speed-dependent. For lack of more detailed

knowledge, the present values are used in this study.

A small comment: We have so far assumed both stiffness and damping to follow Normal distributions. It has been common practice in previous studies to model uncertainties related to soil and aerodynamic damping as Lognormally distributed (sometimes other skewed distributions). However, at a CoV of 0.05 there is almost no difference between the corresponding Normal and Lognormal distributions. Even with a CoV of 0.12, the differences are fairly small. See Fig. 5 for details. Hence,

the impact of this simplification is minor.





### 3.4.3 Turbulence intensity

The turbulence intensity is modeled as Lognormally distributed with a wind speed-dependent mean and a CoV 0.05, i.e. $\theta_{\text{turb}} \sim LN(m, 0.05)$ with $m$ derived from Table 2. This is consistent with, e.g., Sørensen and Tarp-Johansen (2005), Veldkamp (2008) and Toft and Sørensen (2011). The particular value is based on the expected uncertainty in turbulence intensity as derived

from the uncertainty of cup anemometer measurements. If including also the uncertainty from wake modeling in a wind farm, which is not done here, the CoV will be higher (Toft et al., 2016b).

### 3.4.4 Fatigue resistance and yield strength

The fatigue resistance is modeled as Lognormally distributed with a mean of 1.0 and a CoV of 0.3, consistent with e.g. Márquez-Domínguez and Sørensen (2012) and (Toft et al., 2016b), i.e. $\Delta_{\text{F}} \sim LN(-0.431, 0.294)$.

The yield strength is modeled as Lognormally distributed with a mean of 288 MPa and a CoV of 0.1, consistent with e.g. Melchers (1999). To account for some of the effects of the simplifications used to arrive at Eq. (29), the CoV is increased to 0.15. Hence, $f_y \sim LN(5.652, 0.149)$.

The uncertainty modeling is summarized in Table 3.

### 3.5 Implementation details

So far, some of the details of the proposed methodology have been left unspecified in order to suggest an overall framework for RBDO rather than a very specific set of methods. The literature contains a large amount of choice in regards to optimization algorithms (both at the design optimization level and in the inner optimization loop solving the reliability problem), surrogate modeling and DOE. While these details have to be fixed in order to demonstrate the method in practice, the optimal selection of algorithms is not considered within the scope of the present study. Such optimality will in any case be both application specific

and depend on the personal preferences of the designer.

With regards to both levels of the optimization, we use a combination of SQP and interior-point methods (Nocedal and Wright, 2006), both of which are common examples of gradient-based nonlinear constrained optimization algorithms. In principle, the PMA-based reliability problem can be solved more efficiently by use of the hybrid mean-value method (Youn et al., 2003) or related approaches, but due to the use of the surrogate model, this is not deemed necessary for the current application.

**Table 3.** Uncertainty modeling details. Quantities marked with * are expressed relative to the respective deterministic values.

| Parameter | Symbol | Distribution | Mean | CoV |
|---|---|---|---|---|
| Global stiffness | $\theta_{\text{stiff}}$ | Normal | 1.0* | 0.052 |
| Global damping ratio | $\theta_{\text{damp}}$ | Normal | 7.735 | 0.115 |
| Turbulence intensity | $\theta_{\text{turb}}$ | Lognormal | 1.0* | 0.05 |
| Fatigue resistance | $\Delta_{\text{F}}$ | Lognormal | 1.0 | 0.30 |
| Yield strength | $f_y$ | Lognormal | 288 [MPa] | 0.15 |



Convergence of the optimization is based on fairly standard criteria, with termination of the algorithms when either relative first order optimality (see e.g. Nocedal and Wright (2006)) is achieved with a tolerance of $10^{-6}$ or the relative changes in the design variables are less than $10^{-6}$. Solutions are required to be feasible with a tolerance of $10^{-6}$.

For the surrogate modeling, we have chosen GPR due to the benefits stated previously. After some initial trial and error, the Matérn class kernel with $\nu = 3/2$ was chosen, including the use of individual length scale hyperparameters for each input variable (this implements what is known as automatic relevance determination, in principle de-emphasizing less relevant input variables in the regression problem, see e.g. Rasmussen and Williams (2006)). Overall, this was found to be the most robust for the regression problem in this study, especially when considering repeated regression for additional iterations of the outer loop in Algorithm 1. The Matérn class of kernels was also used for OWT support structures in Häfele et al. (2019). We also note here that in order to simplify the simultaneous regression with respect to all of the three parameters in $\theta_q$, these parameters were input to the fitting problem in such a way that the surrogate model became co-monotonic in every variable (an increase in one or more variables giving always an increase in the output). In this case, that meant inputting the inverse ($1/\theta_i$) of the parameters controlling damping and stiffness. Furthermore, these parameters were implemented as scaling variables with means of 1.0, such that the actual variables as input to the simulations were a product of the deterministic values and the respective stochastic scaling parameters in $\theta_q$. The hyperparameters of the Gaussian process model were fit using Bayesian global optimization methods (expected improvement) (Mockus, 1975; Jones et al., 1998; Brochu et al., 2010). The noise standard deviation was taken to be non-zero and also fit during this procedure, even though the simulation outputs used in the fitting are in a certain sense noise-free. This was done because it was seen to give more robust surrogates with respect to changes in the design.

The DOE was done using Sobol sequences, a Quasi-Monte Carlo method. This has the advantage of being more space-filling, covering a larger range of the space while still having some clustering to account for local variations, compared to many ordinary Monte Carlo methods. While not made use of here, Sobol sequences also have the advantage compared to the commonly used Latin hypercube sampling method that it is much easier to interactively add new samples to the old set. To this last point, the use of an adaptive DOE was not used here, despite this increasingly becoming the common approach for GPR. The main reason why was a practical one, having to do with the way the loading input was sampled, which made interactively adding samples during a fitting procedure difficult for our implementation. A total number of 500 samples were pre-generated, with the number of samples actually used increasing for each iteration of the outer loop in the following way: For the initial surrogate model 50 samples were used. The new model at the solution of the first RBDO procedure was then trained with 100 samples and compared with the old model using 25 additional samples, for a total of 125 samples used. All subsequent iterations use the full set of 500 samples, with 400 used for training and 100 for comparing the current model with the previous one. In a sense, the DOE is thus somewhat dynamic, even if it is not adaptive.

Finally, the outer loop needs termination criteria, as indicated in Algorithm 1. One such criterion was chosen to be simply the convergence of the objective function value. Once this value changes less than a certain small tolerance, the outer loop was halted. However, it is possible to terminate slightly earlier if the surrogate model is seen to converge, since in that case the objective function will not change significantly or at all during the next iteration. As implied above, the new surrogate models trained at the solution of the current RBDO loop were thus compared with the models used during that loop. Due to the use




of noisy regression models, the surrogate models will in practice never converge entirely (or will at least do so very slowly) as long as there are small changes in the design (and small changes in the surrogate model give further small changes in the design etc.). Hence, a more relaxed convergence criterion was developed for the surrogate models. Specifically, if we denote by $\bar{y}_{\text{new}}$ and $\bar{y}_{\text{old}}$ the mean prediction of the new and old surrogate models, respectively, and by $\sigma_{y,\text{old}}$ the predicted standard

deviation of the old model, then if

$$|\bar{y}_{\text{new}} - \bar{y}_{\text{old}}| \leq \sigma_{y,\text{old}} \qquad (32)$$

for every test sample point, the surrogate model is not updated. If no surrogate models are updated after an outer iteration, the procedure terminates. In order for this relaxed tolerance not to give infeasible results with respect to the new surrogate model (which is not used when it is within the above tolerance), the surrogate predictions for $y(\theta_q)$, used to compute the numerical

values for the limit state functions in Eq. (30) and Eq. (31), are based on the mean plus the standard deviation, $\bar{y} + \sigma_y$, rather than just the mean. This guarantees that when $\bar{y}_{\text{old}}$ is used instead of the updated model, the derived results remain strictly feasible with respect to mean of the more accurate prediction (which would otherwise have been used). Since the standard deviations tend to be $\sim [10^{-3}, 10^{-2}]$, this does not have a large impact on the results.

## 4   Results

To illustrate both the basic workings of the RBDO method and the effect of certain modeling choices and constraints, a number of different cases are studied. For easy reference, these have been given names and will be referred to as such from now on. The names and properties of each of these cases are listed in Table 4. Note that for cases marked with *Connected*, only one set of values for diameters and thicknesses are used throughout the structure, meaning there are only 2 design variables. In all other cases there is one diameter and thickness per element (giving 12 design variables for the Simple Beam model and

28 for the OC3 Monopile). There is also one set of non-linear constraints (fatigue, extreme load or both) per element. To make comparisons between deterministic and probabilistic optimization more clear, the deterministic non-linear constraint limits have been tuned to match more closely their probabilistic counterparts. In particular, the deterministic versions of the resistance variables $\Delta_{\text{F}}$ and $f_y$ have been set to $H_\theta^{-1}(\Phi(-3.3))$ for $\theta = \theta_{\Delta_{\text{F}}}$ and $\theta = \theta_{f_y}$ respectively. This can be considered a form of simplified safety factor scaling.

### 4.1   Simple Beam

The objective function for case BEAM-PA-CON is shown in panel (a) of Fig. 6. Note how there are only very minor changes after the first loop. The small modifications to the design variables in the second and third loops are caused by the updates in the probabilistic constraints, seen in panel (b) of Fig. 6. The final design is characterized by an overall minimization of thickness while the diameter is increased, as seen in panel (c) of Fig. 6. Comparing with the corresponding deterministic case

BEAM-DA-CON, the main difference is a slightly more conservative design, as would be expected. The corresponding plots





are not shown, as they are almost identical, but results for both cases are summarized in Table 5. Note that the amount of outer iterations for loop 1 of BEAM-PA-CON is about the same as the total number of iterations for BEAM-DA-CON.

The results for BEAM-PA and BEAM-DA are shown in Fig. 7 and Fig. 8 respectively. Compared to the cases with connected design variables, there is an (expected) increase in the number of iterations required to solve the problem and the resulting designs are different in the way that the dimensions are reduced for elements higher in the structure. This is a natural consequence
5   of the fact that the loads are higher towards the bottom and the constraints there will be stricter in terms of allowable cross-sectional dimensions. Otherwise, the results are similar. The non-linear constraints are somewhat closer to being active at the

**Table 4.** Testing cases for RBDO. Loading scenario numbers refer to the values in Table 2

| Case Name | Model | Probabilistic | Loading scenarios | Number of design variables, number of non-linear constraints | Other |
|---|---|---|---|---|---|
| BEAM-PA | Simple Beam | Yes | $1+2+3+4$ | 12, 12 | None |
| BEAM-PA-CON | Simple Beam | Yes | $1+2+3+4$ | 2, 12 | Connected |
| BEAM-DA | Simple Beam | No | $1+2+3+4$ | 12, 12 | None |
| BEAM-DA-CON | Simple Beam | No | $1+2+3+4$ | 2, 12 | Connected |
| OC3-PA | OC3 Monopile | Yes | $1+2+3+4$ | 28, 28 | None |
| OC3-DA | OC3 Monopile | No | $1+2+3+4$ | 28, 28 | None |
| OC3-PF | OC3 Monopile | Yes | $1+2+3$ | 28, 14 | None |
| OC3-PU | OC3 Monopile | Yes | 4 | 28, 14 | None |
| OC3-PA-DT | OC3 Monopile | Yes | $1+2+3+4$ | 28, 28 | D-t-ratio constraint |
| OC3-PA-NW | OC3 Monopile | Yes | $1+2+3+4$ | 28, 28 | No wave loads |
| OC3-PA-RND | OC3 Monopile | Yes | $1+2+3+4$ | 28, 28 | Randomized design |

**Table 5.** Summary results of cases BEAM-PA-CON and BEAM-DA-CON.

| Variable | BEAM-PA-CON | BEAM-DA-CON |
|---|---|---|
| Diameter after loop 1 | 8.98 m | N/A |
| Final diameter | 9.00 m | 8.44 m |
| Thickness after loop 1 | 0.0140 m | N/A |
| Final thickness | 0.0137 m | 0.0133 m |
| Normalized mass after loop 1 | 0.351 | N/A |
| Final normalized mass | 0.346 | 0.315 |
| Initial maximum $P_f$ | $2.0 \cdot 10^{-11}$ | $2.0 \cdot 10^{-11}$ |
| Final maximum $P_f$ | $4.8 \cdot 10^{-4}$ | $1.8 \cdot 10^{-2}$ |
| Iterations in loop 1 | 20 | N/A |
| Total iterations | 36 | 22 |



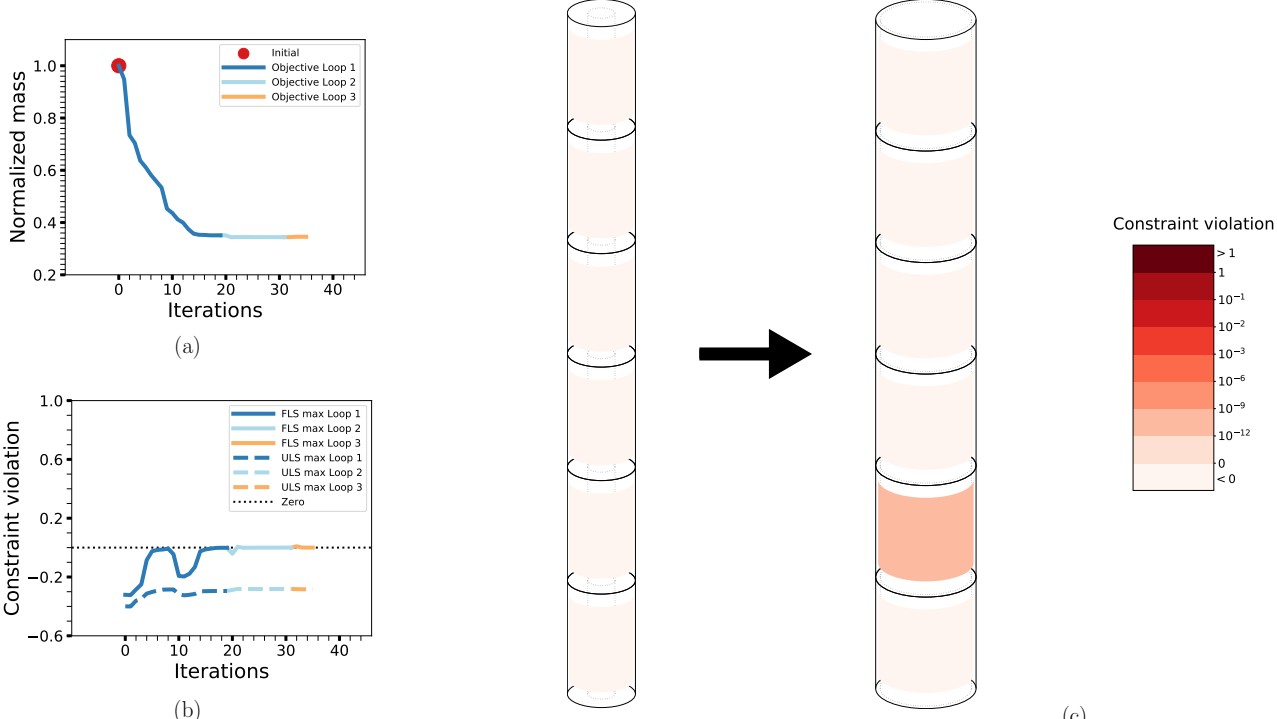

**Figure 6.** The optimization process for case BEAM-PA-CON: The objective function (a), maximum non-linear constraint violations (b) and the change in the design from initial to final (c). The design drawings also have the level of non-linear constraint violation indicated by the coloring of the elements. The thicknesses have been exaggerated for legibility.

solution of the RBDO problem compared to the deterministic case, which was true previously but is more apparent for these cases. Detailed summary results are for these cases displayed in Table 6.

All in all, the results so far show that the method works well for these simple systems. The convergence behavior is more or less as for the deterministic case, with the addition of a few short extra loops to achieve overall convergence with respect to the updated GPR-based surrogate model. However, the results obtained from the first outer loop are likely good enough for practical purposes. The fatigue constraints dominate over the extreme load constraints, which is not unexpected. Furthermore, the system seems driven by the thickness(es) both with respect to the objective (structure mass) and the (fatigue) constraint and the solutions reflect this (with minimal thicknesses and increased diameters where necessary to compensate). This can mostly be understood as a result of the fact that the contribution of the thickness to the cross-sectional areas and second moments of area are of higher order than that of the diameter.

## 4.2 OC3 Monopile

Beginning with the two basic cases for the OC3 Monopile, OC3-PA and OC3-DA, displayed in Fig. 9 and Fig. 10 respectively, we see that the behavior is fairly similar to the Simple Beam cases without connected design variables. Despite having more





than twice the number of design variables, convergence is achieved in about the same number of iterations. The main new detail in the solution is that the second element from the bottom does not follow the otherwise apparent pattern of monotonically increasing diameters from top to bottom. In fact, this element has a smaller diameter than the element above, with a comparatively increased thickness to compensate. This is expected to be due to the wave loads, which are driven more by the diameter. The reason this does not happen for the Simple Beam cases is most likely because the smaller number of elements

can not resolve this effect. Otherwise, there is in the probabilistic case a much larger constraint violation at some intermediate points and the objective function initially increases above its starting value, but this does not seem to have much of an effect on the overall solution. More detailed results are shown in Table 7.

Next, the effect of including D-t-ratio constraints for all elements is shown in the results from case OC3-PA-DT in Fig. 11. With this constraint in place, the low thickness, high diameter solution obtained previously is no longer feasible and the result

is a solution which balances the reduction more evenly among the thicknesses and diameters. The result is in a sense more pleasing from a practical point of view, since it is more in line with a design that would actually be manufactured; both due to the lack of very large diameters and because one avoids the wave load-induced "hourglass shape" seen in the previous two cases. The convergence is also faster (73 vs 140 iterations), though there is one additional (very short) outer iteration required. On the other hand, the D-t-ratio constraint is a lot more strict overall and less than 10 % reduction in mass is possible. In fact,

the initial OC3 design is not feasible with respect to this constraint, which is why the objective function is increased by quite some amount at the beginning of the first loop. Note also that, as opposed to other comparable cases, the final design is softer (at 0.24 Hz) than the initial design (at 0.28 Hz).

**Table 6.** Selected summary results of cases BEAM-PA and BEAM-DA.

| Variable | BEAM-PA | BEAM-DA |
|---|---|---|
| Bottom diameter after loop 1 | 9.00 m | N/A |
| Final bottom diameter | 9.00 m | 9.00 m |
| Bottom thickness after loop 1 | 0.0158 m | N/A |
| Final bottom thickness | 0.0157 m | 0.0144 m |
| Top diameter after loop 1 | 4.17 m | N/A |
| Final top diameter | 4.16 m | 3.95 m |
| Top thickness after loop 1 | 0.0133 m | N/A |
| Final top thickness | 0.0133 m | 0.0133 m |
| Normalized mass after loop 1 | 0.258 | N/A |
| Final normalized mass | 0.259 | 0.238 |
| Initial maximum $P_f$ | $2.0 \cdot 10^{-11}$ | $2.0 \cdot 10^{-11}$ |
| Final maximum $P_f$ | $4.8 \cdot 10^{-4}$ | $2.7 \cdot 10^{-2}$ |
| Iterations in loop 1 | 78 | N/A |
| Total iterations | 128 | 86 |



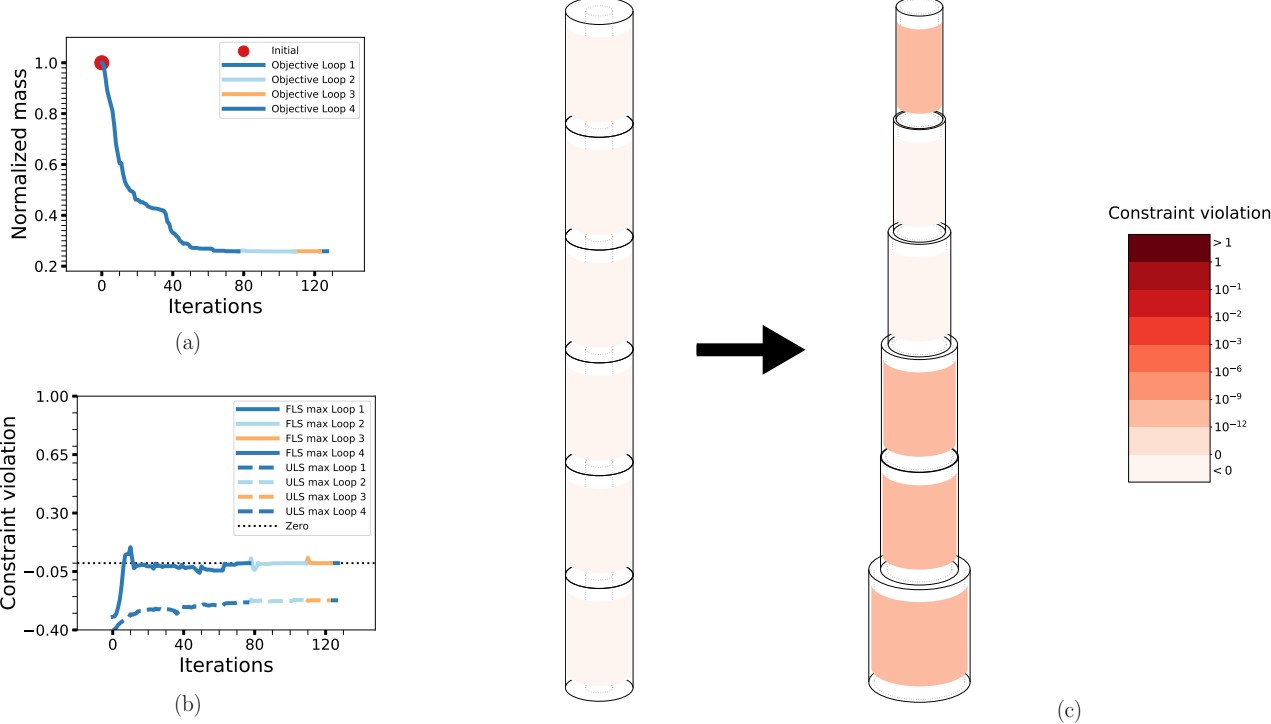

**Figure 7.** The optimization process for case BEAM-PA: The objective function (a), maximum non-linear constraint violations (b) and the change in the design from initial to final (c). Details as in previous figure.

Randomizing the initial OC3 design gives the results displayed for OC3-PA-RND in Fig. 12. This initial design is both heavier and much less feasible (a probability of failure of 1 essentially in several locations) than the initial OC3 design, which seems to make the convergence a bit slower in this case, but not by too much. The solution is not exactly the same as for OC3-PA, but the difference is negligible (1 % or less in the design variables and less than 0.005 % in the objective). This is within the expected variation caused by the small inherent randomness in the surrogate modeling.

Finally, the effects of no wave loads (OC3-PA-NW), only fatigue constraints (OC3-PF) and only extreme load constraints (OC3-PU) are shown in the results in Fig. 13, Fig. 14 and Fig. 15 respectively. As would be expected, the removal of the wave loads leads to a slightly lighter design and one where the element diameters consistently decrease from bottom to top. The difference in convergence behavior is likely negligible and mostly due to the randomness in the surrogate modeling. The resulting design has a slightly higher utilization of extreme loads. Since the fatigue constraints generally dominate over the

extreme load constraints, the results for OC3-PF are as expected, with negligible differences in the final solution compared with OC3-PA and OC3-PA-RND. Conversely, using only extreme load constraints as in OC3-PU results in a final design that has about 16 % less mass than OC3-PA. This case also removes the visible effect of the wave loads on the solution, most likely because (at least for this loading scenario) the extreme loads caused by the waves are much less significant than the corresponding fatigue loads. Otherwise, the behavior is more or less as in the other cases.





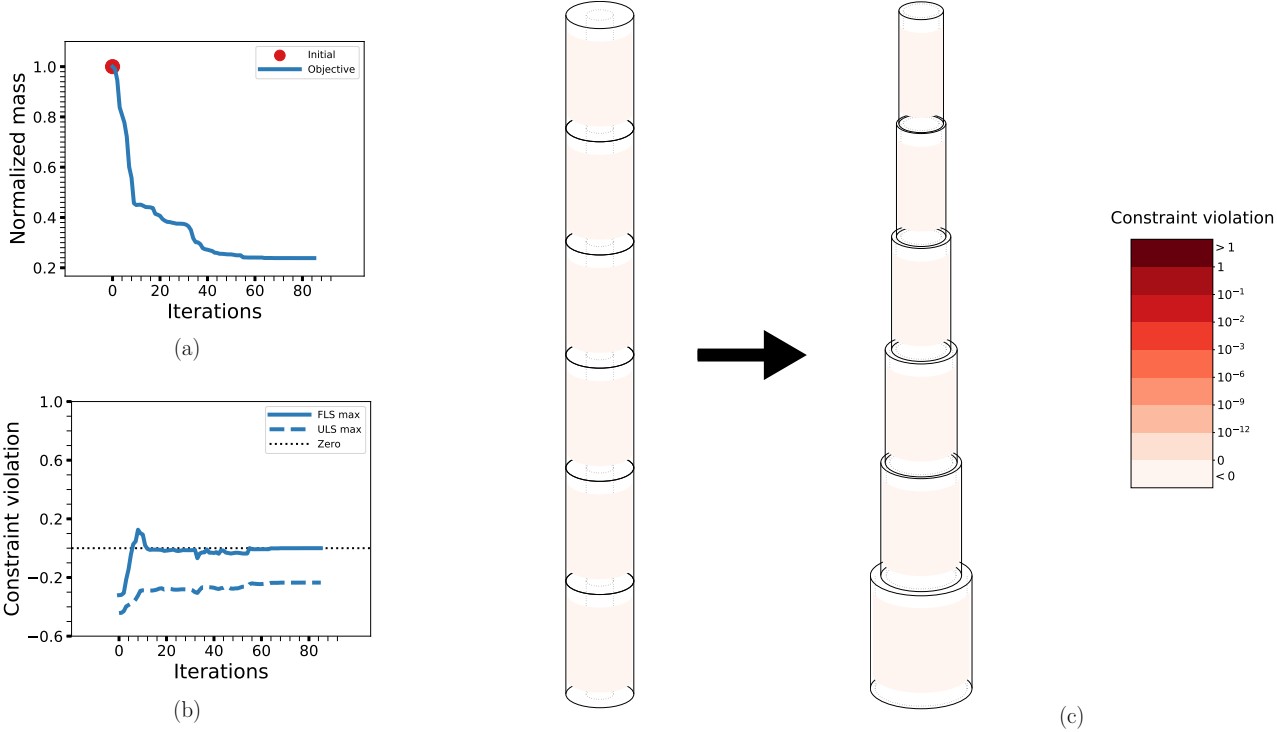

**Figure 8.** The optimization process for case BEAM-DA: The objective function (a), maximum non-linear constraint violations (b) and the change in the design from initial to final (c). Details as in previous figures.

More detailed results for OC3-PA-DT, OC3-PA-RND, OC3-PA-NW, OC3-PF and OC3-PU can be found in Table 8.

## 5    Further discussion

The results demonstrate quite clearly the capability of the proposed methodology to obtain reliable optimal support structure designs without making the optimization process itself much more computationally complex than in the deterministic case. In fact, the initial outer iteration of the RBDO approach require about the same number of iterations as the corresponding

5    deterministic optimization cases. The small amount of changes to the design that occur in the additional outer iterations indicate that, even with the simplifications involved in the response factorization, the surrogate model is a fairly accurate global approximation. Final convergence of the outer loop is then mostly necessary for convergence in a mathematical sense, and the added computational effort required is of lesser practical importance. Tightening the non-linear constraints slightly would ensure that feasible solutions were obtained after only one round of RBDO. The 50 samples used to train the surrogate model

10    for the initial RBDO loop represent a very small additional computational effort compared to what is required for the optimization in general. Since the minimum number of function evaluations (and thus simulations) required for a single iteration is one base evaluation plus one additional evaluation for every design variable, 50 simulations becomes rather negligible (for



the OC3 Monopile models, this number is surpassed after only 2 iterations). Even when using all 500 samples, the added computational effort is not particularly large when compared with a full optimization procedure (it is equivalent to at most 18 iterations of OC3 Monopile models or at most 46 iterations of unconnected Simple Beam models). All in all, this makes the proposed RBDO framework a realistic option if gradient-based deterministic optimization is computationally feasible for the

**Table 7.** Selected summary results of cases OC3-PA and OC3-DA. Design variable numbers run from 1 (bottom element) to 14 (top element).

| Variable | OC3-PA | OC3-DA |
|---|---|---|
| Diameter 1 after loop 1 | 9.00 m | N/A |
| Final diameter 1 | 9.00 m | 9.00 m |
| Thickness 1 after loop 1 | 0.0201 m | N/A |
| Final thickness 1 | 0.0201 m | 0.0179 m |
| Diameter 2 after loop 1 | 6.82 m | N/A |
| Final diameter 2 | 6.84 m | 6.74 m |
| Thickness 2 after loop 1 | 0.0294 m | N/A |
| Final thickness 2 | 0.0295 m | 0.0271 m |
| Diameter 3 after loop 1 | 9.00 m | N/A |
| Final diameter 3 | 9.00 m | 8.76 m |
| Thickness 3 after loop 1 | 0.0139 m | N/A |
| Final thickness 3 | 0.0141 m | 0.0133 m |
| Diameter 4 after loop 1 | 8.52 m | N/A |
| Final diameter 4 | 8.50 m | 8.1 m |
| Thickness 4 after loop 1 | 0.0133 m | N/A |
| Final thickness 4 | 0.0133 m | 0.0133 m |
| Diameter 14 after loop 1 | 4.33 m | N/A |
| Final diameter 14 | 4.35 m | 4.09 m |
| Thickness 14 after loop 1 | 0.0133 m | N/A |
| Final thickness 14 | 0.0133 m | 0.0133 m |
| Normalized mass after loop 1 | 0.588 | N/A |
| Final normalized mass | 0.586 | 0.545 |
| Initial maximum $P_f$ | $7.8 \cdot 10^{-5}$ | $7.8 \cdot 10^{-5}$ |
| Final maximum $P_f$ | $4.8 \cdot 10^{-4}$ | $1.9 \cdot 10^{-2}$ |
| Initial 1st eigenfrequency | 0.279 Hz | 0.279 Hz |
| 1st eigenfrequency after loop 1 | 0.295 Hz | N/A |
| Final 1st eigenfrequency | 0.294 Hz | 0.274 Hz |
| Iterations in loop 1 | 91 | N/A |
| Total iterations | 140 | 85 |





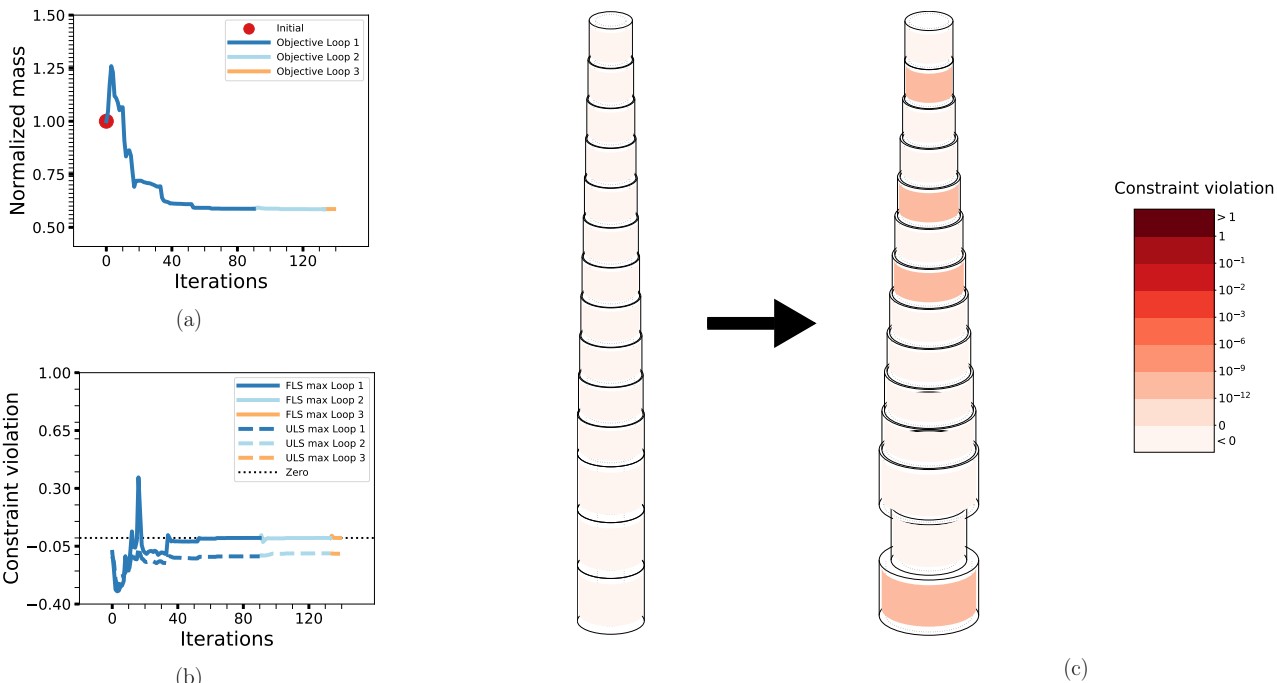

**Figure 9.** The optimization process for case OC3-PA: The objective function (a), maximum non-linear constraint violations (b) and the change in the design from initial to final (c). Details as in previous figures.

desired application. We note that the computation time on a single workstation (16 cores @ 2.7 GHz; 128 GB RAM) for one full evaluation of the constraints (including all simulations required for four loading scenarios and the computation of all design sensitivities) was about 40 seconds for the non-connected Simple Beam designs and about 100 seconds for the OC3 designs. The total solution time was consequently on the order of hours. At most 10-12 hours for all outer iterations to complete, but generally only a few hours for the initial outer iteration.

## 5.1 Obtained designs

The results obtained from RBDO do not appear functionally or systematically different than those obtained with deterministic optimization, producing designs that are similar and only slightly heavier. Note for example the large differences in maximum probability of failure compared to the small differences in total mass. The designs are driven by fatigue on the load side and the element thicknesses on the structural side, leading in general to designs with small thicknesses and large diameters. The OC3 Monopile designs tend to be quite a bit stiffer than the initial design, except when the loads or constraints are relaxed enough to allow for very light designs (as in the case of OC3-PA-NW and OC3-PU). The overall exception to these trends is the case with a D-t-ratio constraint, though the thickness is also driving in this case. However, since the thickness cannot be arbitrarily smaller than the diameters in this case, the result is an effective upper bound on the thickness corresponding to the values where the overall design (with much smaller diameters than the other cases) is at the boundary imposed by the non-linear constraints. All





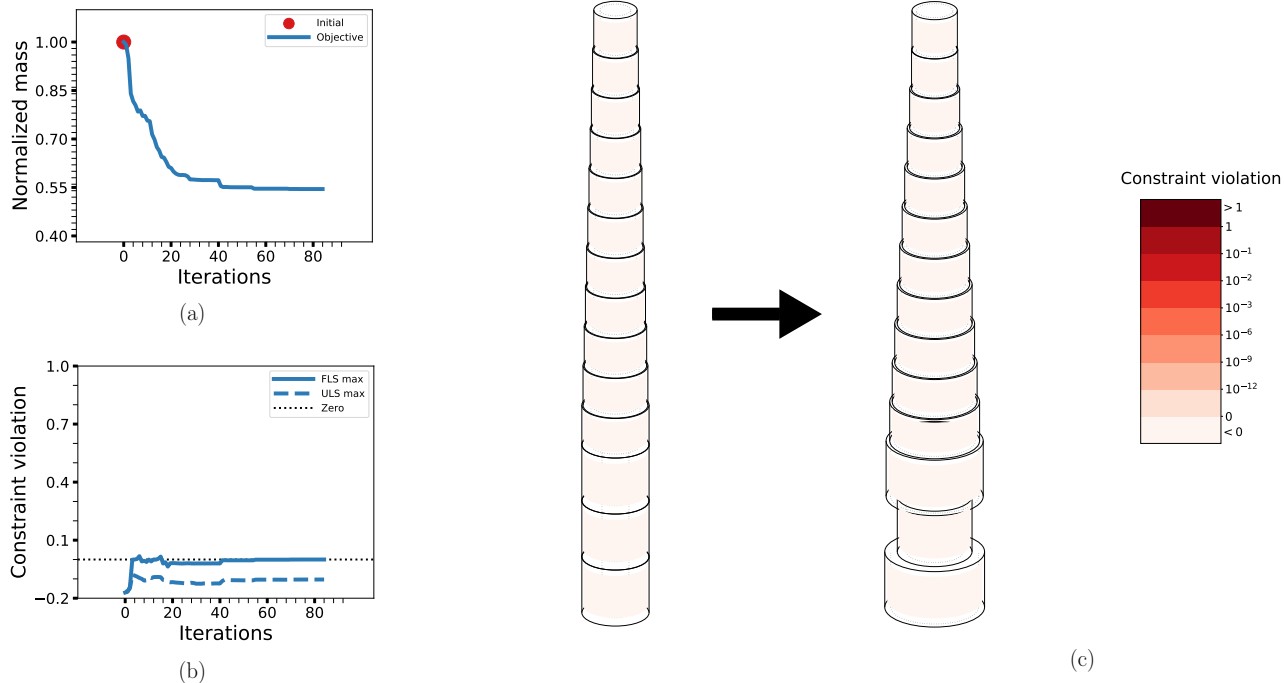

**Figure 10.** The optimization process for case OC3-DA: The objective function (a), maximum non-linear constraint violations (b) and the change in the design from initial to final (c). Details as in previous figures.

in all, this is beneficial for the design process, since reliability-based constraints do not seem to change anything fundamental about the problem or introduce anything phenomenologically new from the design point of view. By and large, this indicates that as long as the structural and load models can be successfully adapted to the probabilistic setting, e.g. in the manner done in this study, then most if not all of previous knowledge and experience from deterministic design optimization is still valid and useful. On the other hand, this should not be taken to mean that probabilistic constraints can be easily replaced by more conservative deterministic ones. There is no way to determine sensible limits for such constraints – sensible here in the sense of being sufficiently safe while not being overly conservative – without performing some kind of non-deterministic analysis. Such analyses, for example probabilistically tuned partial safety factors as basis for deterministically constrained optimization, are not necessarily more efficient than the present RBDO framework and can not account for any potentially design-dependent changes that would be naturally accounted for with our methodology.

## 5.2 Simplifications

Some further simplifications have been made in the present analysis compared with more realistic applications. The main examples are the system model (with no soil model or detailed hydrodynamic modeling), the load analysis (simplified wave modeling, small number of environmental states considered) and the uncertainty modeling (potentially a much larger set of uncertainties might have been considered and a more detailed approach could have been used to obtain the specific uncertainty



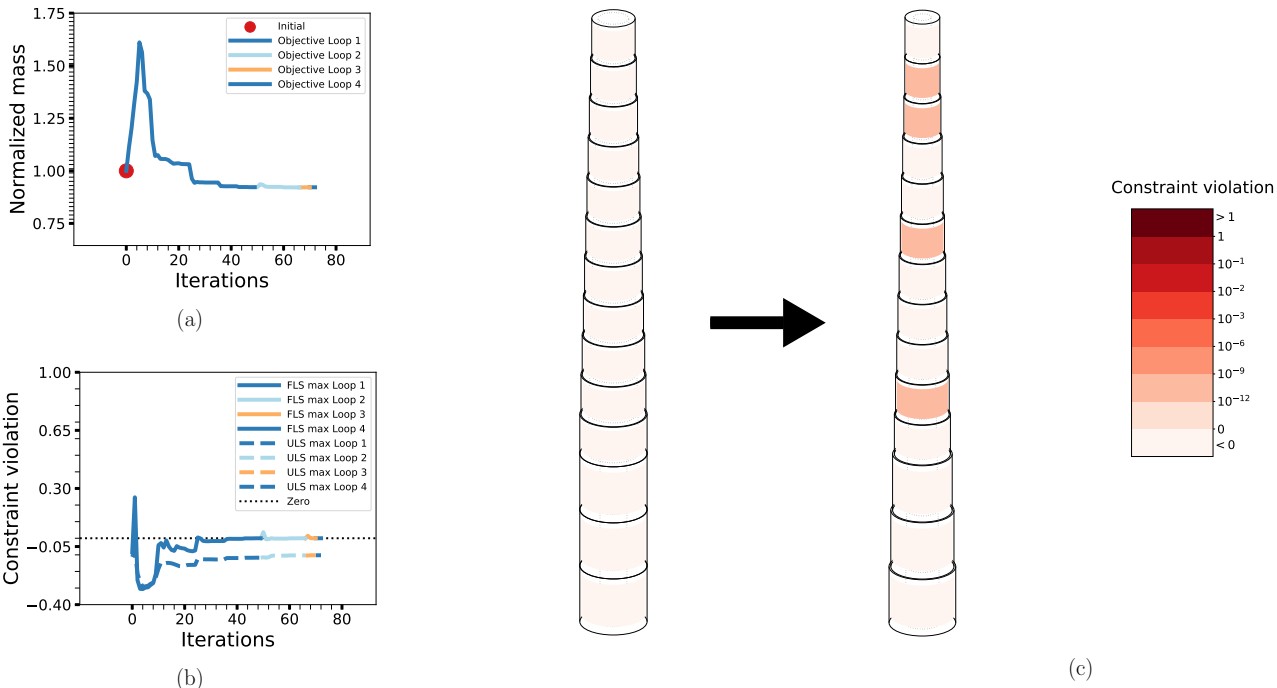

**Figure 11.** The optimization process for case OC3-PA-DT: The objective function (a), maximum non-linear constraint violations (b) and the change in the design from initial to final (c). Details as in previous figures.

15 models). None of these simplifications are negligible, but are not expected to affect the viability of the results dramatically either. The system and load modeling are not necessarily so far away from approaches commonly used for industrial applications, nor do they affect the system response in a way that would cause large deviations from the behavior seen in this study. The simplified (or lack of) soil structure interaction and hydrodynamic properties mostly serve to increase the global stiffness, reduce global damping and change the self weight of the system. These are systematic effects that may change the amplitudes

5 of the response, but are not expected to change the relative response to specific scenarios and so change, e.g., the complexity required to fit the surrogate model with respect to design changes. Similarly, the simplified load analysis is also not expected to affect the relative responses very much. Especially for the fatigue analysis where recent studies have shown that the distribution of fatigue damage over a comprehensive set of environmental states does not change drastically when the design changes, particularly as long as the eigenfrequency does not change too much (Stieng and Muskulus (2018) and Stieng and Muskulus

10 (2019)). Finally, the uncertainty analysis is mostly consistent with previous work in terms of the specific modeling, but uses a smaller number of uncertainties than has typically been part of reliability studies. This can be seen as somewhat limiting in regards to the above points about the lack of added computational complexity, but it should be noted that it is usually possible to reduce the number of uncertainties down to a level closer to that of the present study by careful preliminary studies of the sensitivity to each uncertain parameter and subsequent elimination of all but the most important parameters. The automatic

15 relevance determination of the GPR approach used presently is also advantageous for such a purpose.





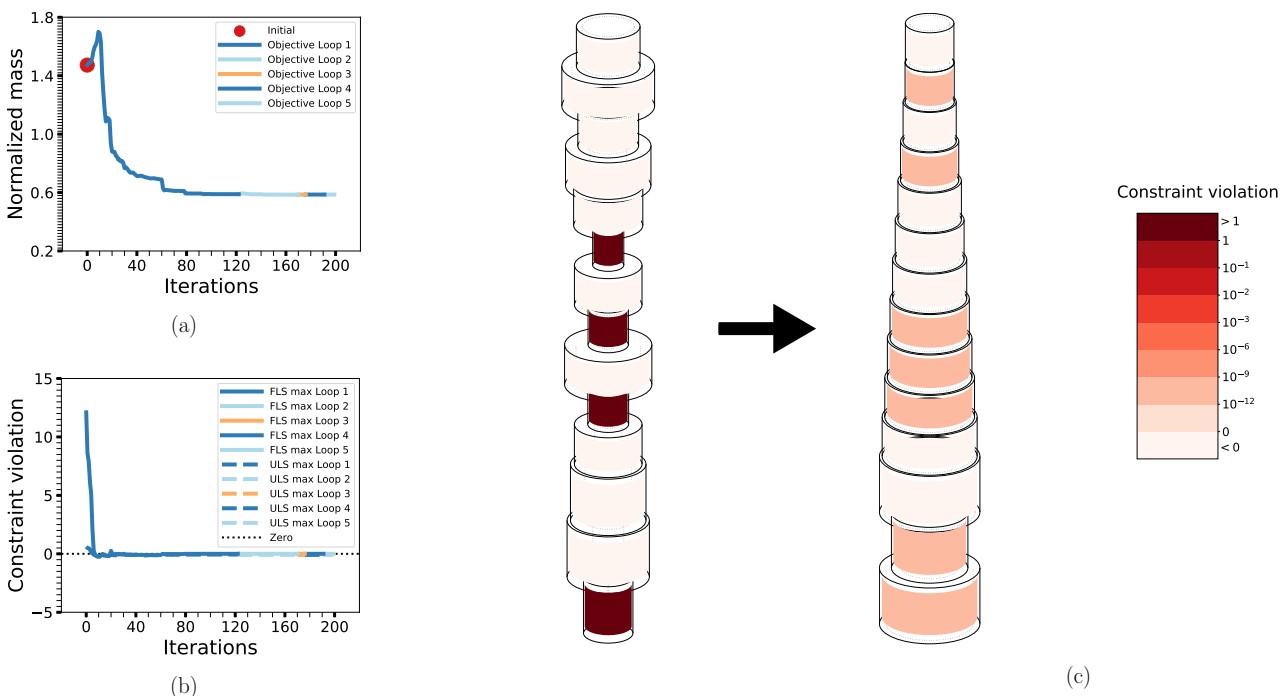

**Figure 12.** The optimization process for case OC3-PA-RND: The objective function (a), maximum non-linear constraint violations (b) and the change in the design from initial to final (c). Details as in previous figures.

## 6 Conclusions

In this work we have presented a general methodology for performing RBDO of OWT support structures. The fundamental idea is that if the stochastic system response can be factorized into a design-dependent, deterministic (mean) response and a design-independent, probabilistic response, then it becomes possible to implement state-of-the-art RBDO, including state-of-the-art support structure design optimization methods, without adding much computational effort compared to deterministic 5 optimization. The further advantages of the approach are that no assumptions about the functional representation of the probabilistic response are necessary and since all design-dependence is found in the deterministic part of the response, high fidelity surrogate models can be fit for the probabilistic response while simultaneously making use of analytical methods for the estimation of design sensitivities. Together, this makes it possible to utilize recently developed gradient-based methods without having to make further adaptations of more general RBDO methods.

10 For the range of considered cases, the results show the feasibility of the proposed methodology. Although the overall approach includes an additional outer loop to ensure local fidelity of the surrogate model at the solution, these additional iterations are only necessary to ensure convergence in a stricter sense. For practical purposes, a single surrogate model fit and a single RBDO procedure suffices. Furthermore, the number of iterations of the RBDO procedure (not counting the solution of each reliability sub-problem, which is computationally negligible when using a surrogate model), and hence the number of simula-





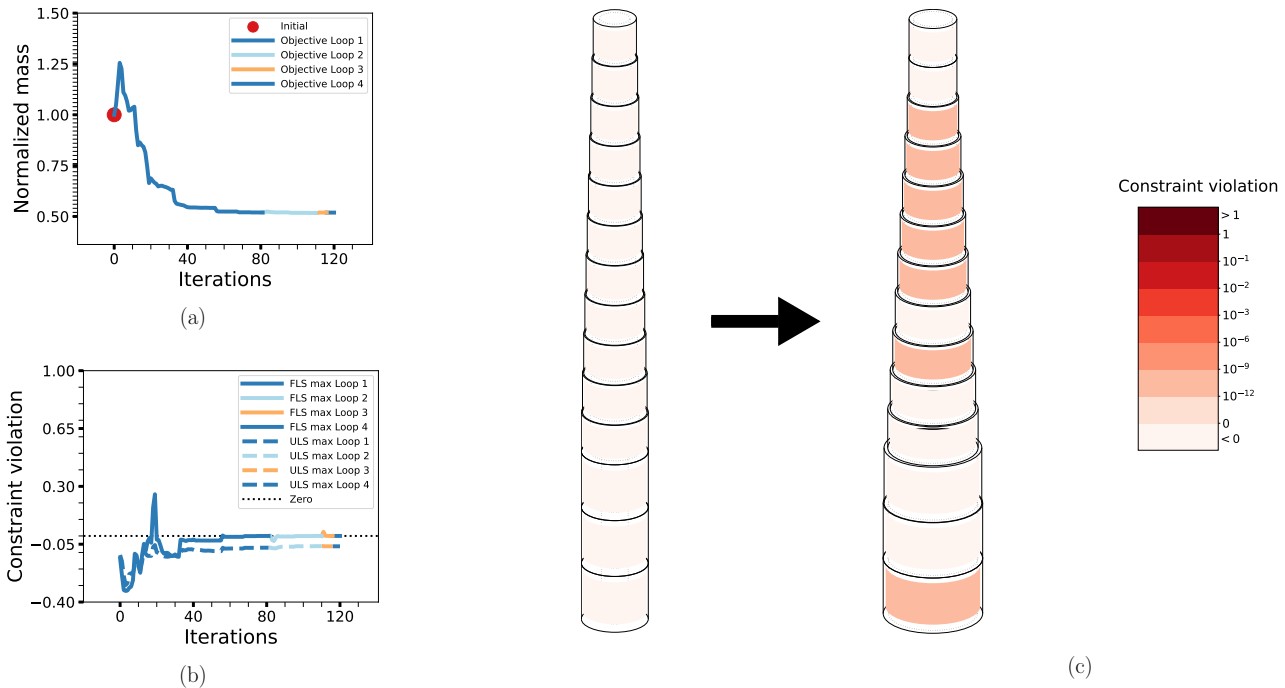

**Figure 13.** The optimization process for case OC3-PA-NW: The objective function (a), maximum non-linear constraint violations (b) and the change in the design from initial to final (c). Details as in previous figures.

tions required during optimization, is very close to that of the equivalent deterministic cases. The only additional computational effort is then found in the training of the surrogate model. However, this effort is comparable to that of a small number of additional iterations of the design optimization, especially for a larger number of design variables. Hence, the overall added computational complexity is small and makes the RBDO problem comparable to the equivalent deterministic optimization problem. The results also indicate that the RBDO framework does not change anything significantly about the kind of optimal designs that are obtained, as compared with deterministic design optimization. The same properties (fatigue and element thickness) seem to drive the designs and the main differences are that probabilistically constrained designs are more conservative than their deterministic counterparts, as one would expect.

The current study is somewhat preliminary, in the sense that only a limited number of loading scenarios and constraints are considered, as well as the fact that the structural and environmental models are simplified and that limited effort has been put into refining, or otherwise optimizing, the methods used in the implementation of the overall framework. With regards to the simplifications, this is not expected to be a very limiting factor, though future work with higher fidelity is needed to ensure the practical viability of the proposed approach. As for the lack of refinement, this would indicate at least some potential for improving the methodology presented herein, which already works fairly well. It is likely that at the very least a more efficient design of experiment will be crucial if a larger amount of loading scenarios and higher fidelity system modeling is to be made practical. Considering that many of the underlying optimization procedures used were originally developed for



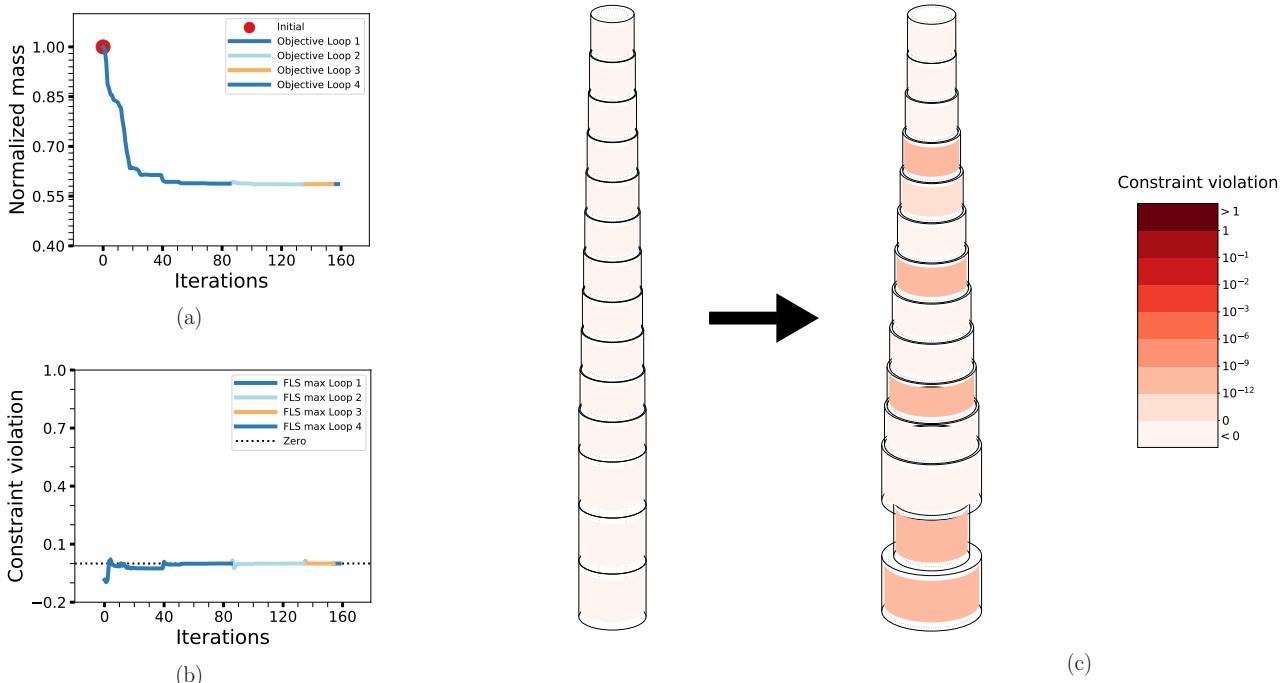

**Figure 14.** The optimization process for case OC3-PF: The objective function (a), maximum non-linear constraint violations (b) and the change in the design from initial to final (c). Details as in previous figures.

jacket support structures, it is expected that the current results, derived for monopiles, should be applicable with only minor modifications. Since very few studies of RBDO for OWTs have been done so far, in particular for support structure design, the current developments will hopefully open up new avenues for further research.

*Code and data availability.* The data used for creating the figures and tables displaying the results is available as supplementary material. The code used to generate the results is very comprehensive and is in its current form not suitable for publication.

5 *Author contributions.* Lars Einar S. Stieng formulated the main idea and implemented the method, conducted the analysis, created the figures and wrote the manuscript. Michael Muskulus provided essential input and suggestions throughout the process, aided in the formulation of the scope of the work and helped with the composition of the manuscript.

*Competing interests.* The authors declare that they have no competing interests.



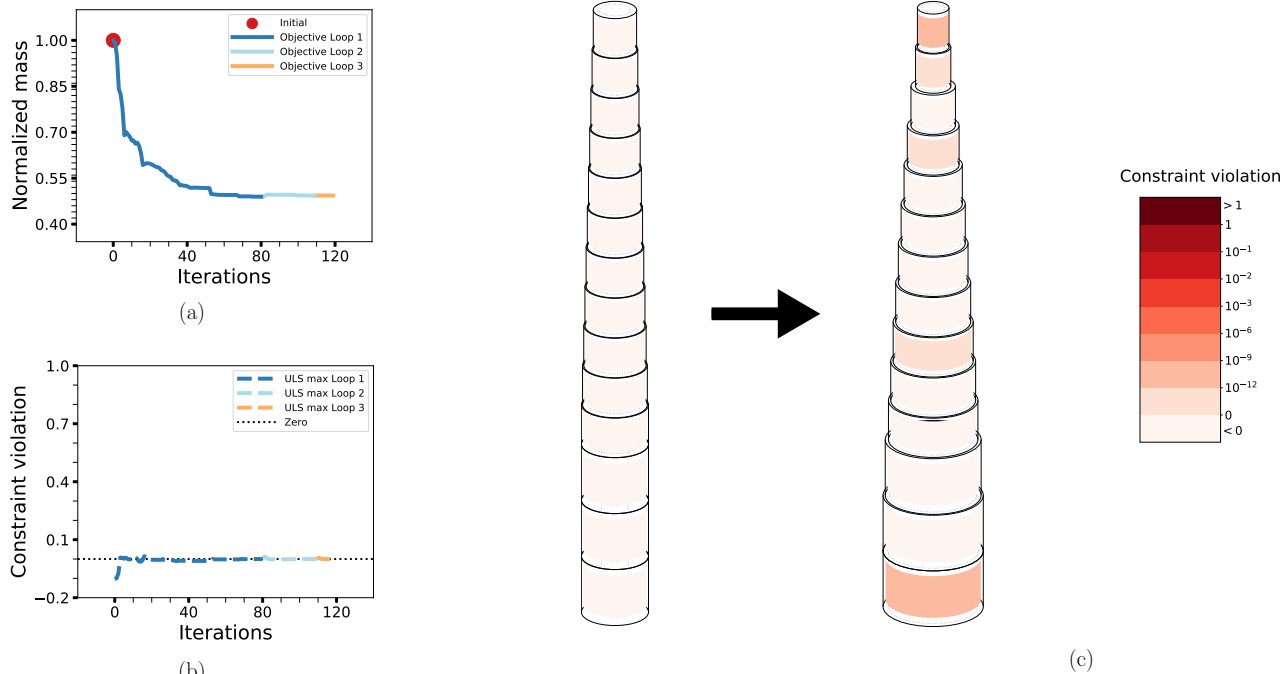

**Figure 15.** The optimization process for case OC3-PU: The objective function (a), maximum non-linear constraint violations (b) and the change in the design from initial to final (c). Details as in previous figures.

*Acknowledgements.* This work has been partly supported by NOWITECH FME (Research Council of Norway, contract no. 193823) and by the Danish Council for Strategic Research through the project "Advancing BeYond Shallow waterS (ABYSS) - Optimal design of offshore wind turbine support structures".





**Table 8.** Selected summary results of cases OC3-PA-DT, OC3-PA-RND, OC3-PA-NW, OC3-PF and OC3-PU. Design variable numbers run from 1 (bottom element) to 14 (top element).

| Variable | OC3-PA-DT | OC3-PA-RND | OC3-PA-NW | OC3-PF | OC3-PU |
|---|---|---|---|---|---|
| Diameter 1 after loop 1 | 5.90 m | 9.00 m | 9.00 m | 9.00 m | 9.00 m |
| Final diameter 1 | 5.92 m | 9.00 m | 9.00 m | 9.00 m | 9.00 m |
| Thickness 1 after loop 1 | 0.0491 m | 0.0198 m | 0.0164 m | 0.0202 m | 0.0163 m |
| Final thickness 1 | 0.0493 m | 0.0202 m | 0.0165 m | 0.0202 m | 0.0165 m |
| Diameter 2 after loop 1 | 5.58 m | 6.81 m | 9.00 m | 6.82 m | 9.00 m |
| Final diameter 2 | 5.61 m | 6.85 m | 9.00 m | 6.85 m | 9.00m |
| Thickness 2 after loop 1 | 0.0465 m | 0.0145 m | 0.0145 m | 0.0294 m | 0.0141 m |
| Final thickness 2 | 0.0467 m | 0.0294 m | 0.0147 m | 0.0294 m | 0.0143 m |
| Diameter 3 after loop 1 | 5.26 m | 9.00 m | 8.81 m | 9.00 m | 8.54 m |
| Final diameter 3 | 5.29 m | 9.00 m | 8.84 m | 9.00 m | 8.61 m |
| Thickness 3 after loop 1 | 0.0438 m | 0.0142 m | 0.0133 m | 0.0139 m | 0.0133 m |
| Final thickness 3 | 0.0441 m | 0.0141 m | 0.0133 m | 0.0141 m | 0.0133 m |
| Diameter 4 after loop 1 | 4.97 m | 8.70 m | 8.20 m | 8.52 m | 7.98 m |
| Final diameter 4 | 4.98 m | 8.51 m | 8.18 m | 8.50 m | 8.03 m |
| Thickness 4 after loop 1 | 0.0414 m | 0.0133 m | 0.0133 m | 0.0133 m | 0.0133 m |
| Final thickness 4 | 0.0415 m | 0.0133 m | 0.0133 m | 0.0133 m | 0.0133 m |
| Diameter 14 after loop 1 | 3.09 m | 4.33 m | 4.31 m | 4.33 m | 2.84 m |
| Final diameter 14 | 3.09 m | 4.35 m | 4.32 m | 4.35 m | 2.84 m |
| Thickness 14 after loop 1 | 0.0257 m | 0.0133 m | 0.0133 m | 0.0133 m | 0.0133 m |
| Final thickness 14 | 0.0258 m | 0.0133 m | 0.0133 m | 0.0133 m | 0.0133 m |
| Normalized mass after loop 1 | 0.922 | 0.589 | 0.519 | 0.588 | 0.490 |
| Final normalized mass | 0.922 | 0.586 | 0.519 | 0.586 | 0.493 |
| Initial 1st eigenfrequency | 0.279 Hz | 0.217 Hz | 0.279 Hz | 0.279 Hz | 0.279 Hz |
| 1st eigenfrequency after loop 1 | 0.236 Hz | 0.296 Hz | 0.283 Hz | 0.295 Hz | 0.266 Hz |
| Final 1st eigenfrequency | 0.235 Hz | 0.294 Hz | 0.282 Hz | 0.294 Hz | 0.268 Hz |
| Iterations in loop 1 | 50 | 123 | 82 | 86 | 81 |
| Total iterations | 73 | 200 | 121 | 160 | 120 |





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
