# Peer review of "Reliability-based design optimization of offshore wind turbine support structures using analytical sensitivities and factorized uncertainty modeling"

_Wind Energy Science, 2019_

## Referee Comment (RC1) · Anonymous Referee #1 · 23 Sep 2019

The paper is of high quality, well structured. It demonstrates a methodology for efficient reliability-based optimization of offshore wind turbine support structures (applied to monopile structures), including uncertainty aspects together with the design optimization.

General comments:

The paper is well written with high-qualitative formulations. The paper is also well structured, however, the reviewer suggests to add a paragraph at the end of the introduction

section to introduce the structure of the paper.

Specific comments:

- At several points, the gradient-based and gradient-free approaches and differences are discussed. The reviewer suggests to mention directly within the abstract why specifically gradient-based design optimization is addressed and applied within the approach demonstrated in this paper. The benefit of gradient-based methods over gradient-free methods is mentioned just on page 6 (lines 3 and 4) - this should be mentioned already at an earlier point in the paper. Furthermore, the argumentation and presentation of the shortcomings of gradient-based methods, mentioned in lines 12-15 on page 6, brings up again the question why not gradient-free methods are used, if gradient-based methods are faster converging, but might not converge at all or present inaccurate solutions. Thus, the argumentation for the decision to use gradient-based methods in this approach should be clearer and more straightforward.

- In the introduction section (lines 12 and 13 on page 2), the main distinction between robust and reliability-based design optimization is highlighted, however, a short explanation what the differences are is missing.

- Please provide numbers to support your comparisons in the introduction (e.g. for lines 19-21 on page 3).

- Missing details:

o Which finite element tool is used (mentioned in section 3 on page 16)?

o For the constraints of the diameters and thicknesses the specific values (70% and 150%) are mentioned based on manufacturing/transportation/installation constraints as well as simulation constraints. However, 150% * 6m = 9m is no manufacturing/transportation/installation constraint. The constraint for ill-behaved simulations is not defined in more detail. Thus, the reviewer recommends to include a table, presenting the limits for the practical (manufacturing/transportation/installation) and finite
element constraints (simulation feasibility), so that it is clear to the reader where the 70% and 150% bounds come from.

o For the constraints upper bounds on the accumulated 20-year fatigue damage and on the maximum bending moment are mentioned in section 3.2 (lines 11 and 12 on page 18), however, no values or any information on how these bounds are derived are stated.

o For the additional constraints, presented on page 20, equations with further parameters are presented and used. Some values for some parameters are discussed and indicated, however, several values are not specified (e.g. the constants $a_i$, the used Wöhler exponents $w_i$, the applied reference thickness $t\_ref$ with corresponding thickness correction exponent k, the selected fatigue resistance $Delta\_F$, as well as the constant r for controlling the accuracy of the approximation).

- In section 3.1 on page 17 the models and loads are introduced. However, the author should present more clearly, if the externally calculated loads are determined for each geometry anew. Based on the descriptions in section 3.1 the question arises, what happens with diameter-dependent loads, when the design is changed, especially in the not-connected case, as a tapered structure or a structure with jumps in the diameter has other load effects than a straight cylinder. Based on the descriptions within the example on page 28 (lines 3-5), it seems that the loads are calculated for each geometry investigated within the optimization. This fact should be mentioned clearly in section 3.1.
* * *

---

## Referee Comment (RC2) · Anonymous Referee #2 · 24 Oct 2019

This paper presents an efficient methodology for reliability-based design optimisation by decoupling the reliability analysis from the design optimisation. The methodology is applied to several different cases based on a uniform cantilever beam and the OC3 monopile and different loading and constraints scenarios. The results have demonstrated the viability of the proposed method.

Specific comments are as follows.

1. Introduction: It would be appropriate to include a paragraph to review the available

optimisation algorithms and justify the choice of gradient-based optimisation used in this study.

2. Methodology: It would be appropriate to add a flowchart of the proposed framework for RBDO of OWT support structures.

For the constraints, please justify why other constraints, such as buckling and vibration (frequency), are not considered in this study.

3. Testing and implementation details: Lack of case studies to validate the key components of the RBDO framework, e.g. the finite element model.

Part of the OC3 monopile is actually embedded into the soil. The soil-structure interaction can significantly affect the structural performance of the monopile. Please justify why the soil is not considered in this study.

Please clarify how the loads were applied, and clarify if the wave loads are updated with the change of diameters during the optimisation process.

4. Results: Results are presented well.

5. Further discussion: Informative discussion is presented.

---

## Author Comment (AC1) · 7 Nov 2019

**Author response to interactive comments on "Reliability-based design optimization of offshore wind turbine support structures using analytical sensitivities and factorized uncertainty modeling"**

Lars Einar S. Stieng          Michael Muskulus

We would like to begin by thanking both referees for their careful reading of the manuscript and their thoughtful and helpful comments. This feedback will serve to improve the quality of the manuscript considerably.

Below, we will respond to the comments of Referee 1 first and then Referee 2 second. In each case, the comments from the respective referee will be italicized and our response for each comments follows in normal type face.

**Response to Referee #1**

*The paper is of high quality, well structured. It demonstrates a methodology for efficient reliability-based optimization of offshore wind turbine support structures (applied to monopile structures), including uncertainty aspects together with the design optimization.*

We once again thank the referee for their work on reviewing the paper.

**General:**

*The paper is well written with high-qualitative formulations. The paper is also well structured, however, the reviewer suggests to add a paragraph at the end of the introduction section to introduce the structure of the paper*

Since the paper is fairly long and comprehensive, this is a good suggestion and will be included in the revised manuscript.

**Specific:**

*At several points, the gradient-based and gradient-free approaches and differences are discussed. The reviewer suggests to mention directly within the abstract why specifically gradient-based design optimization is addressed and applied within the approach demonstrated in this paper.*

A comment regarding the efficiency of gradient-based methods will be added to the abstract in the revised manuscript.

*The benefit of gradient-based methods over gradient-free methods is mentioned just on page 6 (lines 3 and 4) - this should be mentioned already at an earlier point in the paper. Furthermore, the argumentation and presentation of the shortcomings of gradient-based methods, mentioned in lines 12-15 on page 6, brings up again the question why not gradient-free methods are used, if gradient-based methods are faster converging, but might not converge at all or present inaccurate solutions. Thus, the argumentation for the decision to use gradient-based methods in this approach should be clearer and more straightforward*

A comment on the motivation for using gradient-based methods will be added to the relevant discussion in the introduction. Furthermore, the highlighted discussion on page 6 will be clarified to underline that the efficiency of gradient-based methods makes these preferable to gradient-free methods, especially when analytical sensitivities can remove the problems with accuracy and/or lack of convergence. The revised manuscript will reflect this.

*In the introduction section (lines 12 and 13 on page 2), the main distinction between robust and reliability-based design optimization is highlighted, however, a short explanation what the differences are is missing.*

We agree that a sentence or two quickly underlining the qualitative differences between the two approaches, rather than just implying there is one, would be useful for the reader. This will be added in the revised manuscript.

*Please provide numbers to support your comparisons in the introduction (e.g. for lines 19-21 on page 3).*

Specific numbers will be added to the highlighted part in the revised manuscript in order to make this point more clear.

**Missing details:**

*Which finite element tool is used (mentioned in section 3 on page 16)*

An in-house finite element code written in MATLAB was used, this will be mentioned in the revised manuscript.

*For the constraints of the diameters and thicknesses the specific values (70% and 150%) are mentioned based on manufacturing/transportation/installation constraints as well as simulation constraints. However, 150% \* 6m = 9m is no manufacturing/transportation/installation constraint. The constraint for ill-behaved simulations is not defined in more detail. Thus, the reviewer recommends to include a table, presenting the limits for the practical (manufacturing/transportation/installation) and finite element constraints (simulation feasibility), so that it is clear to the reader where the 70% and 150% bounds come from.*

These constraints are not defined according to strict criteria. Rather, the inclusion of upper and lower bounds in general results from the idea of manufacturing constraints and the specific bounds chosen have been set by hand for this case, as it was observed during early testing of the code that the numerical behavior outside of these bounds was less stable. This should have been underlined more clearly in the text and a more clear explanation will be included in the revised manuscript.

*For the constraints upper bounds on the accumulated 20-year fatigue damage and on the maximum bending moment are mentioned in section 3.2 (lines 11 and 12 on page 18), however, no values or any information on how these bounds are derived are stated.*

This was perhaps ill phrased in the text. By "upper bounds", what we were referring to were the constraints represented by Eq. (27) and Eq. (29), which later are also transformed into probabilistic constraints with given limits. The terminology "upper bounds" is confusing in this context and another phrasing should have been used to make the meaning more clear. This will be updated in the revised manuscript.

*For the additional constraints, presented on page 20, equations with further parameters are presented and used. Some values for some parameters are discussed and indicated, however, several values are not specified (e.g. the constants $a_i$, the used Wöhler exponents $w_i$, the applied reference thickness $t_{ref}$ with corresponding thickness correction exponent $k$, the selected fatigue resistance $\Delta_F$, as well as the constant $r$ for controlling the accuracy of the approximation).*

These constants will be defined/have stated values in the revised manuscript.

*In section 3.1 on page 17 the models and loads are introduced. However, the author should present more clearly, if the externally calculated loads are determined for each geometry anew. Based on the descriptions in section 3.1 the question arises, what happens with diameter-dependent loads, when the design is changed, especially in the not-connected case, as a tapered structure or a structure with jumps in the diameter has other load effects than a straight cylinder. Based on the descriptions within the example on page 28 (lines 3-5), it seems that the loads are calculated for each geometry investigated within the optimization. This fact should be mentioned clearly in section 3.1.*

The rotor loads applied at tower top have been extracted from fixed rotor (with no tower or support structure) simulations and so are only generated once per environmental state, but the dynamic response obtained when these loads are applied to our model is calculated for each new design. The wave loads depend on the diameter of the lower parts of the monopile through the Morison equation and are updated for each new design before the

estimation of the dynamic response. Both these points will be underlined clearly in the revised manuscript.

**Response to Referee #2**

*This paper presents an efficient methodology for reliability-based design optimisation by decoupling the reliability analysis from the design optimisation. The methodology is applied to several different cases based on a uniform cantilever beam and the OC3 monopile and different loading and constraints scenarios. The results have demonstrated the viability of the proposed method.*

We once again thank the referee for their work on reviewing the paper.

*Introduction: It would be appropriate to include a paragraph to review the available optimisation algorithms and justify the choice of gradient-based optimisation used in this study.*

A short review of optimization approaches relevant for support structures would indeed be instructive for the reader and will be added to the introduction in the revised manuscript. Further justification of gradient-based optimization will be added in accordance with the previous response to Referee 1.

*Methodology: It would be appropriate to add a flowchart of the proposed framework for RBDO of OWT support structures.*

This was already suggested by Algorithm 1, but an additional flowchart would probably be helpful for readers. Such a figure will be added in the revised manuscript.

*For the constraints, please justify why other constraints, such as buckling and vibration (frequency), are not considered in this study*

This was mainly to make the study more focused on the probabilistic aspect, rather than the specific structural analysis. A more realistic study would need to implement such constraints, but since our main focus is testing the presented ideas, the selected constraint types are regarded as sufficient/illustrative for this purpose. Having a smaller number of constraints also makes it easier to isolate what is going on in terms of how these probabilistic constraints affect the optimization. An explanation of this will be added to the revised manuscript.

*Testing and implementation details: Lack of case studies to validate the key components of the RBDO framework, e.g. the finite element model.*

While some readers might wish for such a validation, it is a bit outside the scope of the paper, which is already very comprehensive in other respects and focuses more specifically on optimization and reliability analysis, to consider the finite element modeling in such detail. The finite element model is not intended for further development or for more realistic application, so the effort required for a thorough validation would not lead to

comparable benefits. On the other hand, some amount of validation may be obtained by observing the first eigenfrequency as reported in (e.g.) Table 7, which is more or less consistent with results in other studies with the OC3 Monopile and NREL 5MW turbine and otherwise similar setups.

*Part of the OC3 monopile is actually embedded into the soil. The soil-structure interaction can significantly affect the structural performance of the monopile. Please justify why the soil is not considered in this study.*

This is again a matter of simplification. While the presence of the soil does indeed affect the loads on the structure, which is why soil effects have been included indirectly as part of the studied uncertainty, this effect is generally systematic (and somewhat predictable) when compared to a structure fixed at the seabed. Hence, while the soil interaction is important for structural performance, it does not have a direct consequence on the performance of the more general RBDO-method and so, to simplify the modeling, the soil was not included. A clarification of this will be added to the revised manuscript.

*Please clarify how the loads were applied, and clarify if the wave loads are updated with the change of diameters during the optimisation process*

The application of the loads were explained briefly in Section 3.1, but this could have been more clear and will be updated in the revised manuscript. A clarification about wave loads being updated when diameters change will be added, as noted in a previous response comment to Referee 1, in the revised manuscript.

---

## Author Response (AR1)

**Author response to interactive comments on "Reliability-based design optimization of offshore wind turbine support structures using analytical sensitivities and factorized uncertainty modeling"**

Lars Einar S. Stieng        Michael Muskulus

We would like to begin by thanking both referees for their careful reading of the manuscript and their thoughtful and helpful comments. This feedback will serve to improve the quality of the manuscript considerably.

**Please note that a version of the revised manuscript with changes indicated in blue, for additions, and red, for deletions, has been attached at the end of this document.**

Below, we will respond to the comments of Referee 1 first and then Referee 2 second. In each case, the comments from the respective referee will be italicized and our response for each comments follows in normal type face. References to specific changes in the new manuscript, with locations indicated, are written in bold type face.

**Response to Referee #1**

*The paper is of high quality, well structured. It demonstrates a methodology for efficient reliability-based optimization of offshore wind turbine support structures (applied to monopile structures), including uncertainty aspects together with the design optimization.*

We once again thank the referee for their work on reviewing the paper.

**General:**

*The paper is well written with high-qualitative formulations. The paper is also well structured, however, the reviewer suggests to add a paragraph at the end of the introduction section to introduce the structure of the paper*

Since the paper is fairly long and comprehensive, this is a good suggestion.

A paragraph indicating the structure of the paper was added to the introduction in the revised manuscript: Last paragraph of page 5 in both the revised manuscript and the marked up version.

**Specific:**

*At several points, the gradient-based and gradient-free approaches and differences are discussed. The reviewer suggests to mention directly within the abstract why specifically gradient-based design optimization is addressed and applied within the approach demonstrated in this paper.*

**A comment regarding the efficiency of gradient-based methods was added to the abstract in the revised manuscript: Page 1, lines 5-7 in both the revised manuscript and the marked up version.**

*The benefit of gradient-based methods over gradient-free methods is mentioned just on page 6 (lines 3 and 4) - this should be mentioned already at an earlier point in the paper. Furthermore, the argumentation and presentation of the shortcomings of gradient-based methods, mentioned in lines 12-15 on page 6, brings up again the question why not gradient-free methods are used, if gradient-based methods are faster converging, but might not converge at all or present inaccurate solutions. Thus, the argumentation for the decision to use gradient-based methods in this approach should be clearer and more straightforward*

**A comment on the motivation for using gradient-based methods was added to the relevant discussion in the introduction: Page 4, lines 5-10 in the revised manuscript; page 4, lines 6-11 in the marked up version.**

Furthermore, the highlighted discussion on page 6 will be clarified to underline that the efficiency of gradient-based methods makes these preferable to gradient-free methods, especially when analytical sensitivities can remove the problems with accuracy and/or lack of convergence.

**This discussion was modified accordingly: Page 7, lines 6-15 in the revised manuscript; page 7, lines 8-17 in the marked up version.**

*In the introduction section (lines 12 and 13 on page 2), the main distinction between robust and reliability-based design optimization is highlighted, however, a short explanation what the differences are is missing.*

We agree that a sentence or two quickly underlining the qualitative differences between the two approaches, rather than just implying there is one, would be useful for the reader.

**A short explanation regarding this was added to the introduction: Page 2, lines 14-17 in both the revised manuscript and the marked up version.**

*Please provide numbers to support your comparisons in the introduction (e.g. for lines*

*19-21 on page 3).*

**Specific numbers were added to the highlighted part in the revised manuscript in order to make this point more clear: Page 3, lines 24 and 26 in both the revised manuscript and the marked up version.**

**Missing details:**

*Which finite element tool is used (mentioned in section 3 on page 16)*

An in-house finite element code written in MATLAB was used.

**This was mentioned in the revised manuscript: Page 17, line 21 in both the revised manuscript and the marked up version.**

*For the constraints of the diameters and thicknesses the specific values (70% and 150%) are mentioned based on manufacturing/transportation/installation constraints as well as simulation constraints. However, 150% * 6m = 9m is no manufacturing/transportation/installation constraint. The constraint for ill-behaved simulations is not defined in more detail. Thus, the reviewer recommends to include a table, presenting the limits for the practical (manufacturing/transportation/installation) and finite element constraints (simulation feasibility), so that it is clear to the reader where the 70% and 150% bounds come from.*

These constraints are not defined according to strict criteria. Rather, the inclusion of upper and lower bounds in general results from the idea of manufacturing constraints and the specific bounds chosen have been set by hand for this case, as it was observed during early testing of the code that the numerical behavior outside of these bounds was less stable. This should have been underlined more clearly in the text.

**A more clear explanation of this was added to the revised manuscript: Page 19, line 12 to page 21, line 6 in the revised manuscript; Page 19, line 12 to page 21, line 8 in the marked up version.**

*For the constraints upper bounds on the accumulated 20-year fatigue damage and on the maximum bending moment are mentioned in section 3.2 (lines 11 and 12 on page 18), however, no values or any information on how these bounds are derived are stated.*

This was perhaps ill phrased in the text. By "upper bounds", what we were referring to were the constraints represented by Eq. (27) and Eq. (29), which later are also transformed into probabilistic constraints with given limits. The terminology "upper bounds" is confusing in this context and another phrasing should have been used to make the meaning more clear.

**This was updated accordingly in the revised manuscript: Page 21, lines 8-9 in the revised manuscript; page 21, lines 9-10 in the marked up version.**

*For the additional constraints, presented on page 20, equations with further parameters are presented and used. Some values for some parameters are discussed and indicated,*

however, several values are not specified (e.g. the constants $a_i$, the used Wöhler exponents $w_i$, the applied reference thickness $t_{ref}$ with corresponding thickness correction exponent $k$, the selected fatigue resistance $\Delta_F$, as well as the constant $r$ for controlling the accuracy of the approximation).

**These constants were defined and/or given stated values in the revised manuscript: Page 22, lines 6-7, 12 and 23 in the revised manuscript; page 22, lines 8-9, 14 and 25 in the marked up version.**

*In section 3.1 on page 17 the models and loads are introduced. However, the author should present more clearly, if the externally calculated loads are determined for each geometry anew. Based on the descriptions in section 3.1 the question arises, what happens with diameter-dependent loads, when the design is changed, especially in the not-connected case, as a tapered structure or a structure with jumps in the diameter has other load effects than a straight cylinder. Based on the descriptions within the example on page 28 (lines 3-5), it seems that the loads are calculated for each geometry investigated within the optimization. This fact should be mentioned clearly in section 3.1.*

The rotor loads applied at tower top have been extracted from fixed rotor (with no tower or support structure) simulations and so are only generated once per environmental state, but the dynamic response obtained when these loads are applied to our model is calculated for each new design. The wave loads depend on the diameter of the lower parts of the monopile through the Morison equation and are updated for each new design before the estimation of the dynamic response.

**Both these points were underlined clearly in the revised manuscript: Page 18, lines 13-14 and page 19, lines 4-5 in the revised manuscript in both the revised manuscript and the marked up version.**

**Response to Referee #2**

*This paper presents an efficient methodology for reliability-based design optimisation by decoupling the reliability analysis from the design optimisation. The methodology is applied to several different cases based on a uniform cantilever beam and the OC3 monopile and different loading and constraints scenarios. The results have demonstrated the viability of the proposed method.*

We once again thank the referee for their work on reviewing the paper.

*Introduction: It would be appropriate to include a paragraph to review the available optimisation algorithms and justify the choice of gradient-based optimisation used in this study.*

A short review of optimization approaches relevant for support structures would indeed be instructive for the reader.

**This was added to the introduction in the revised manuscript: Page 4, lines**

**1-5 in both the revised manuscript and the marked up version.**

**Further justification of gradient-based optimization was added to the introduction in accordance with the previous response to Referee 1.**

*Methodology: It would be appropriate to add a flowchart of the proposed framework for RBDO of OWT support structures.*

This was already suggested by Algorithm 1, but an additional flowchart would probably be helpful for readers.

**A flowchart was added as Figure 4 in the revised manuscript: Page 19 in both the revised manuscript and the marked up version.**

*For the constraints, please justify why other constraints, such as buckling and vibration (frequency), are not considered in this study*

This was mainly to make the study more focused on the probabilistic aspect, rather than the specific structural analysis. A more realistic study would need to implement such constraints, but since our main focus is testing the presented ideas, the selected constraint types are regarded as sufficient/illustrative for this purpose. Having a smaller number of constraints also makes it easier to isolate what is going on in terms of how these probabilistic constraints affect the optimization.

**An explanation of this was added to the revised manuscript: Page 23, lines 3-6 in the revised manuscript; page 23, lines 6-9 in the marked up version.**

*Testing and implementation details: Lack of case studies to validate the key components of the RBDO framework, e.g. the finite element model.*

While some readers might wish for such a validation, it is a bit outside the scope of the paper, which is already very comprehensive in other respects and focuses more specifically on optimization and reliability analysis, to consider the finite element modeling in such detail. The finite element model is not intended for further development or for more realistic application, so the effort required for a thorough validation would not lead to comparable benefits. On the other hand, some amount of validation may be obtained by observing the first eigenfrequency as reported in (e.g.) Table 7, which is more or less consistent with results in other studies with the OC3 Monopile and NREL 5MW turbine and otherwise similar setups.

*Part of the OC3 monopile is actually embedded into the soil. The soil-structure interaction can significantly affect the structural performance of the monopile. Please justify why the soil is not considered in this study.*

This is again a matter of simplification. While the presence of the soil does indeed affect the loads on the structure, which is why soil effects have been included indirectly as part of the studied uncertainty, this effect is generally systematic (and somewhat predictable) when compared to a structure fixed at the seabed. Hence, while the soil interaction is important for structural performance, it does not have a direct consequence on the

performance of the more general RBDO-method and so, to simplify the modeling, the soil was not included.

**A clarification of this issue was added to the revised manuscript: Page 18, lines 9-11 in both the revised manuscript and the marked up version.**

*Please clarify how the loads were applied, and clarify if the wave loads are updated with the change of diameters during the optimisation process*

The application of the loads were explained briefly in Section 3.1, but this could have been more clear.

**This was explained more clearly in the revised manuscript: Page 18, lines 14-16 and page 18, lines 17-18 to page 19, line 1 in both the revised manuscript and the marked up version.**

**A clarification about wave loads being updated when diameters change was added, as noted in a previous response comment to Referee 1, in the revised manuscript.**

**Attachment: Marked up version of revised manuscript**

[revised manuscript text omitted]